# An Error Analysis of Flow Matching for Deep Generative Modeling

**Zhengyu Zhou** [1]  **Weiwei Liu** [1]

## Abstract

Continuous Normalizing Flows (CNFs) have proven to be a highly efficient technique for generative modeling of complex data since the introduction of Flow Matching (FM). The core of FM is to learn the constructed velocity fields of CNFs through deep least squares regression. Despite its empirical effectiveness, theoretical investigations of FM remain limited. In this paper, we present the first end-to-end error analysis of CNFs built upon FM. Our analysis shows that for general target distributions with bounded support, the generated distribution of FM is guaranteed to converge to the target distribution in the sense of the Wasserstein-2 distance. Furthermore, the convergence rate is significantly improved under an additional mild Lipschitz condition of the target score function.

## 1. Introduction

Contemporary generative models have primarily been designed around the construction of a map between two probability distributions that transform samples from the prior distribution to the target distribution. The roots of transport-based sampling and density estimation can be traced back to maximum entropy methods for Gaussianizing data (Tabak & Turner, 2013; Tabak & Vanden-Eijnden, 2010). Normalizing Flows (NFs) provide a neural network implementation of these methods by imposing a structured transformation to make the change of measure tractable in discrete, sequential steps (Dinh et al., 2017; Durkan et al., 2019; Huang et al., 2018; Papamakarios et al., 2017; Rezende & Mohamed, 2015). Continuous Normalizing Flows (CNFs) extend this idea to a continuous-time setting by viewing the map $T(x) = X_t(x)$ as the solution of an ordinary differential equation (ODE) (Chen et al., 2018; Grathwohl et al., 2019). However, training neural ODEs at scale is intractable, as it requires simulating the ODE. The introduction of Flow Matching (FM) has made CNFs highly efficient for generative modeling of complex data (Karras et al., 2022; Liu et al., 2023; Albergo & Vanden-Eijnden, 2023; Lipman et al., 2023; Neklyudov et al., 2022; Tong et al., 2023; Chen & Lipman, 2023; Albergo et al., 2023; Shi et al., 2023; De Bortoli et al., 2021).

The success of FM motivates a line of research investigating the generation quality guarantees from the perspective of sampling (Albergo & Vanden-Eijnden, 2023; Albergo et al., 2023; Lu et al., 2022; Chen et al., 2023c). These works assume the underlying velocity field is accurately estimated up to a small error under $L^2$-norm and provide generation quality guarantees. However, two issues remain unsolved in these works. The first is to provide guarantees for learning the velocity field of the underlying ODE. The second is to relax the strong assumptions on the underlying velocity field, which may be hard to check. This paper takes a step forward by providing an end-to-end analysis[1] of the deep generative modeling based on FM under mild assumptions. Our main contributions are summarized as follows:

- We provide the first end-to-end analysis for the deep generative models based on FM.

- We prove that the deep generative models built upon FM are guaranteed to converge to the target distribution under *mild* assumption. Furthermore, the convergence rate gets significantly *improved* under an additional Lipschitz condition of the target score function.

### 1.1. Assumptions

**Assumption 1.1** (Bounded support)**.** The target distribution $\pi_1$ is supported on $[0,1]^d$.

**Assumption 1.2** (Lipschitz score)**.** Let $\pi_1(\mathrm{d}\boldsymbol{x}) =$

---

[1]School of Computer Science, National Engineering Research Center for Multimedia Software, Institute of Artificial Intelligence and Hubei Key Laboratory of Multimedia and Network Communication Engineering, Wuhan University, Wuhan, China. Correspondence to: Weiwei Liu <liuweiwei863@gmail.com>.

*Proceedings of the 42nd International Conference on Machine Learning*, Vancouver, Canada. PMLR 267, 2025. Copyright 2025 by the author(s).

[1]End-to-end learning in generative modeling involves using finite samples from the target distribution as input to learn the underlying distribution, and then generating samples from the learned distribution as output. The goal of end-to-end analysis is to provide guarantees for the accuracy of the learned distribution based on the finite input samples, enabling more reliable generative modeling.

$e^{-V(\boldsymbol{x})}\mathrm{d}\boldsymbol{x}$. Moreover, the potential $V(\boldsymbol{x})$ is twice continuously differentiable and satisfies $-\alpha I \preceq \nabla^2 V(\boldsymbol{x}) \preceq \alpha I$ with $\alpha > 1$.

**Lemma 1.3.** *Suppose that Assumption 1.1 holds. Then $\boldsymbol{v}^*(\boldsymbol{x}, t)$ is $\xi$-Lipschitz continuous w.r.t. $\boldsymbol{x}$ on $\mathbb{R}^d \times [0, T]$, where $\xi \leq \max\left\{\frac{1}{1-T}, \frac{Td}{(1-T)^3}\right\}$. Further, if $\frac{1}{2} < T < 1$, we have $\boldsymbol{v}^*$ is $\frac{d}{(1-T)^3}$-Lipschitz continuous w.r.t. $\boldsymbol{x}$.*

**Lemma 1.4.** *Suppose that Assumption 1.1 and Assumption 1.2 hold. Then $\boldsymbol{v}^*(\boldsymbol{x}, t)$ is $\zeta(\alpha, d)$-Lipschitz continuous on $\mathbb{R}^d \times [0, 1]$ w.r.t. $\boldsymbol{x}$, where $\zeta(\alpha, d) = \frac{d}{2}\left(\alpha + \sqrt{\alpha + \frac{2}{d}}\right)^2$ scales polynomially with $\alpha$ and $d$.*

*Remark* 1.5. Previous work simply assumes the score function or velocity field to be Lipschitz continuous w.r.t. $\boldsymbol{x}$ for every $t$ (Chen et al., 2023c;a). In this paper, we follow Wibisono & Jog (2018a;b); Mikulincer & Shenfeld (2021; 2022); Chewi & Pooladian (2022); Gao et al. (2024) to provide the Lipschitz continuity of the velocity field from the properties of the target distribution.

The proofs in this section are deferred to Appendix D.5.

## 1.2. Main Results

All proofs of this section is deferred to Appendix C.3.

**Theorem 1.6** (Consistency)**.** *Suppose Assumption 1.1 holds. Given $n$ samples from target distribution $\pi_1$ and the networks as in Theorem 4.4, with parameter $\zeta$ replaced by $\frac{d}{(1-T)^3}$, we use the estimated velocity field in (11), to generate samples and choose the maximal step size $\max_{k=0,1\ldots,N-1}|t_{k+1} - t_k| = \mathcal{O}(n^{-\frac{1}{d+5}})$ and early stopping time $T(n) = 1 - (\log n)^{-1/6}$, we have*

$$W_2(\widetilde{\pi}_{T(n)}, \pi_1) \to 0, \quad \text{in probability,}$$

*where $\widetilde{\pi}_{T(n)}$ denotes the generated distribution at time $T(n)$.*

The consistency of FM is mainly based on a mild assumption, i.e. boundedness, which justifies the use of CNFs based on FM.

**Theorem 1.7** (Improved convergence rate)**.** *Suppose Assumption 1.1 and Assumption 1.2 hold. Given $n$ samples from target distribution $\pi_1$ and the networks as in Theorem 4.4, with parameter $\zeta$ replaced by $\zeta(\alpha, d)$ defined in Lemma 1.4, we use the estimated velocity field in (11) to generate samples and choose the maximal step size $\max_{k=0,1\ldots,N-1}|t_{k+1} - t_k| = \mathcal{O}(n^{-\frac{4}{3(d+5)}})$ and early stopping time $T(n) = 1 - n^{-\frac{1}{3(d+5)}}$. Then, with probability of at least $1 - \frac{1}{n}$, we have*

$$W_2(\widetilde{\pi}_{T(n)}, \pi_1) = \widetilde{\mathcal{O}}\left(n^{-\frac{1}{3(d+5)}}\right),$$

*where $\widetilde{\pi}_{T(n)}$ denotes the generated distribution at time $T(n)$.*

This result highlights the effectiveness of CNFs based on FM in learning the underlying smooth distribution.

## 1.3. Related Work

**Continuous Normalizing Flows** CNFs are proposed by viewing the map $T(x) = X_t(x)$ as the solution of an ODE. It is not until the introduction of FM that CNFs have grown to be an efficient method for the generative modeling of complex data (Karras et al., 2022; Liu et al., 2023; Albergo & Vanden-Eijnden, 2023; Lipman et al., 2023; Neklyudov et al., 2022; Tong et al., 2023; Chen & Lipman, 2023; Albergo et al., 2023; Shi et al., 2023; De Bortoli et al., 2021). The key idea of FM is to learn the constructed velocity fields of CNFs through deep least squares regression. In (Liu et al., 2023), a linear interpolant is proposed with a focus on straight paths. This is employed as a step towards rectifying the transport paths (Liu, 2022) through a procedure which improves sampling efficiency. In (Lipman et al., 2023), the interpolant picture is assembled from the perspective of conditional probability paths connecting to a Gaussian, where a noise convolution is used to improve the learning, at the cost of biasing the method. The paper (Tong et al., 2023) introduces a novel simulation-free objective for learning continuous-time flows conditioned on a general distribution. Further, the authors have shown that lifting the static optimal transport problem to the dynamic setting leads to more efficient training and inference of flow models by lowering the variance of the objective and simplifying flows. FM is extended to the Riemannian setting by Chen & Lipman (2023). Another line of work points out that the probability path of CNFs encompasses that of the Diffusion Models (DMs) (Albergo et al., 2023; Lipman et al., 2023; Albergo & Vanden-Eijnden, 2023). If made to match the performance of their stochastic counterparts, ODE-based methods exhibit a number of desirable characteristics that are absent for SDEs, such as an exact, computationally tractable formula for the likelihood and easy application of well-developed adaptive integration schemes for sampling. Further, one of the most successful techniques of accelerating continuous time process-based sampling, distillation (Liu et al., 2023; Song et al., 2023; Salimans & Ho, 2022; Zheng et al., 2022; Luhman & Luhman, 2021), requires deterministic samplers.

**Lipschitz Score v.s. Lipschitz Velocity Field** In analyzing the convergence of DMs and ODE-based models, the assumption of Lipschitz continuity for the score function or the velocity field has been widely used in previous works (Chen et al., 2023c; Lu et al., 2022; Albergo & Vanden-Eijnden, 2023; Chen et al., 2023a). However, these works simply assume the Lipschitzness. In contrast, our paper takes a step forward and rigorously proves that the velocity field is Lipschitz continuous under mild assumptions on the target distribution. By doing so, we provide a stronger

theoretical foundation for the application of CNFs based on FM, and help to bridge the gap between theory and practice.

**Analysis of ODE-based Models** Significant recent works (Albergo et al., 2023; Chen et al., 2023c; Lu et al., 2022) have put effort into controlling the KL divergence between the generated distribution and the target distribution. These studies have demonstrated that simply regressing the velocity field is insufficient to control the likelihood with ODE-based models. Instead, more advanced learning schemes are required to ensure that the Fisher divergence is kept under control. The work (Albergo & Vanden-Eijnden, 2023) has shown that the Wasserstein-2 distance between the generated distribution and the target distribution can be controlled by the objective of regressing the velocity field, assuming the estimated velocity field is Lipschitz continuous. In our paper, we take a different approach, demonstrating that the true velocity field can be well approximated by a Lipschitz neural network. We compare our work with concurrent analyses for ODE-based models in Table 1 where $U(t; \delta_1, \delta_2, \delta_3, C, q)$ in the third row is an increasing function for $\delta_1, \delta_2$ and $\delta_3$ (Lu et al., 2022), where $\delta_i$ is an upper bound for the score matching objective of order $i$, $i = 1, 2, 3$.

## 2. Preliminaries

**Notations** We denote $[N] := \{0, \cdots, N-1\}$. For matrix $A$ and $B$, we say $A \preceq B$, if $B - A$ is positive semi-definite. We denote the identity matrix in $\mathbb{R}^{d \times d}$ by $I_d$. For a vector $\boldsymbol{x} \in \mathbb{R}^d$, we define $\boldsymbol{x}^{\otimes 2} := \boldsymbol{x}\boldsymbol{x}^T$. We denote the $\ell^2$-norm of a vector $\boldsymbol{x}$ by $\|\boldsymbol{x}\| := \sqrt{\sum_{i=1}^d x_i^2}$. We define the operator norm of a matrix A as $\|A\|_{\mathrm{op}} := \sup_{\|\boldsymbol{x}\| \leq 1} \|A\boldsymbol{x}\|$. For a twice continuously differentiable function $f : \mathbb{R}^d \to \mathbb{R}$, let $\nabla f, \nabla^2 f$, and $\Delta f$ denote its gradient, Hessian, and Laplacian, respectively. For a probability density function $\pi$ and a measurable function $f : \mathbb{R}^d \to \mathbb{R}$, we define the $L^2(\pi)$-norm of $f$ as $\|f\|_{L^2(\pi)} := \left(\int (f(\boldsymbol{x}))^2 \pi(\boldsymbol{x})\mathrm{d}\boldsymbol{x}\right)^{1/2}$. We define $L^\infty(K)$-norm as $\|f\|_{L^\infty(K)} := \sup_{\boldsymbol{x} \in K} |f(\boldsymbol{x})|$. For a vector function $\boldsymbol{v} : \mathbb{R}^d \to \mathbb{R}^d$, we define its $L^2(\pi)$-norm as $\|\boldsymbol{v}\|_{L^2(\pi)} := \|\|\boldsymbol{v}\|\|_{L^2(\pi)}$ and $L^\infty(K)$-norm as $\|\boldsymbol{v}\|_{L^\infty(K)} := \|\|\boldsymbol{v}\|\|_{L^\infty(K)}$. We use the asymptotic notation $f(x) = \mathcal{O}(g(x))$ to denote the statement that $f(x) \leq Cg(x)$ for some constant $C > 0$ and $\widetilde{\mathcal{O}}(\cdot)$ to ignore the logarithm. Given two distributions $\mu$ and $\nu$, the Wasserstein-2 distance is defined as $W_2(\mu, \nu) := \inf_{\pi \in \Pi(\mu,\nu)} \mathbb{E}_{(x,y) \sim \pi}[\|x - y\|^2]^{1/2}$, where $\Pi(\mu, \nu)$ is the set of all couplings of $\mu$ and $\nu$. A coupling is a joint distribution on $\mathbb{R}^d \times \mathbb{R}^d$ whose marginals are $\mu$ and $\nu$ on first and second factors, respectively.

**Flow Matching** Given independent empirical observations of $X_0 \sim \pi_0$ and $X_1 \sim \pi_1$, we want to find an ordinary

differential equation (ODE) on time $t \in [0, 1]$,

$$\mathrm{d}Z_t = \boldsymbol{v}(Z_t, t)\mathrm{d}t, \tag{1}$$

which converts $Z_0$ from $\pi_0$ to $Z_1$ following $\pi_1$. A line of research (Liu et al., 2023; Liu, 2022; Albergo & Vanden-Eijnden, 2023; Lipman et al., 2023; Neklyudov et al., 2022; Wu et al., 2022; Lee et al., 2023b; Tong et al., 2023; Chen & Lipman, 2023; Albergo et al., 2023; Shi et al., 2023) points out that, the vector field can be found by solving a least square regression problem:

$$\min_{\boldsymbol{v}} \mathcal{L}_0(\boldsymbol{v}) := \int_0^1 \mathbb{E}_{X_0, X_1} \left[ \|(X_1 - X_0) - \boldsymbol{v}(X_t, t)\|^2 \right] \mathrm{d}t,$$
$$\text{with} \quad X_t = tX_1 + (1 - t)X_0, \tag{2}$$

where $X_0 \sim \pi_0$, $X_1 \sim \pi_1$, and $X_t$ is the linear interpolation between $X_0$ and $X_1$. The exact minimum of (2) is achieved by

$$\boldsymbol{v}^*(\boldsymbol{x}, t) = \mathbb{E}[X_1 - X_0 | X_t = \boldsymbol{x}]. \tag{3}$$

**Velocity Field Approximation** In practice, the velocity field $\boldsymbol{v}^*$ is approximated by neural networks. To avoid instability, we often clip the integral interval $[0, 1]$ with $T$. Namely, we consider the following loss function:

$$\min_{\boldsymbol{v}} \mathcal{L}(\boldsymbol{v}) := \frac{1}{T} \int_0^T \mathbb{E}_{X_0, X_1} \left[ \|(X_1 - X_0) - \boldsymbol{v}(X_t, t)\|^2 \right] \mathrm{d}t,$$
$$\text{with} \quad X_t = tX_1 + (1 - t)X_0, \tag{4}$$

Given a family of neural networks NN, we consider the following approximation error,

$$\inf_{\boldsymbol{v} \in \mathrm{NN}} \int_0^T \|\boldsymbol{v}(\cdot, t) - \boldsymbol{v}^*(\cdot, t)\|_{L^2(\pi_t)}^2 \mathrm{d}t = \inf_{\boldsymbol{v} \in \mathrm{NN}} \mathcal{L}(\boldsymbol{v}) - \mathcal{L}(\boldsymbol{v}^*), \tag{5}$$

where $\pi_t$ is the probability distribution of $X_t$ defined in (2). The equivalence in (5) is deferred to Lemma 4.1. We also consider the best approximator in the neural networks

$$\widetilde{\boldsymbol{v}} \in \operatorname*{argmin}_{\boldsymbol{v} \in \mathrm{NN}} \mathcal{L}(\boldsymbol{v}). \tag{6}$$

We organize the remaining sections as follows: In Section 3, we show that the true velocity field can be well approximated by a Lipschitz neural network. Section 4 establishes that the optimal neural network can be efficiently estimated. Finally, in Section 5, we analyze the error of distribution recovery using the estimated velocity field.

## 3. Approximation

In practice, the true velocity field is approximated by neural networks. To ensure effective learning, the network class should be expressive enough to approximate the true velocity field.

| | Main Assumptions | End-to-end Analysis | Theoretical Results |
|---|---|---|---|
| (Albergo & Vanden-Eijnden, 2023) | $\hat{\boldsymbol{v}}$ is $\widehat{K}$-Lipschitz in $\boldsymbol{x}$ uniformly on $(t, \boldsymbol{x}) \in [0, 1] \times \mathbb{R}^d$ | ✗ | $W_2^2(\rho_1, \widehat{\rho}_1) \leq e^{1+2\widehat{K}} H(\hat{\boldsymbol{v}})$ |
| (Chen et al., 2023c) | $\nabla \ln q_t^{\leftarrow}(x)$ is $L_{\text{sc},t}$-Lipschitz in $x$ and satisfies $\|\nabla \ln \frac{q_t^{\leftarrow}}{q_s^{\leftarrow}}(x)\| \leq \beta|t-s|^c(1 + \|x\| + \|\nabla q_t^{\leftarrow}(x)\|)$ | ✗ | $\text{KL}(\widehat{p}\|q) \leq \epsilon$ provided $\ell \geq \mathfrak{C}_1$ and $\ell h \leq \mathfrak{C}_2^{-1}$, where $\mathfrak{C}_1$ and $\mathfrak{C}_2$ depends polynomially on parameters in assumptions |
| (Lu et al., 2022) | $\|\nabla_{\boldsymbol{x}}^2 \log p^{\text{ODE}}(\boldsymbol{x}_t)\|_2 \leq C$, $\nabla \log q_t$ is $C$-Lipschitz, uniformly for $t$ | ✗ | $D_F(q_t\|p_t^{\text{ODE}}) \leq U(t; \delta_1, \delta_2, \delta_3, C, q)$ |
| Ours | Bounded support | ✓ | Consistency |
| | Bounded support and Lipschitzness of the target score functions | ✓ | $W_2(\widetilde{\pi}_{T(n)}, \pi_1) = \widetilde{\mathcal{O}}\left(n^{-\frac{1}{3(d+5)}}\right)$ |

*Table 1.* Comparison of existing theoretical results on ODE-based models.

**Neural Network Structure** We configure the ReLU network $\boldsymbol{v}_\theta$ in the following way.

$$\text{NN}(L, M, J, K, \kappa, \gamma_1, \gamma_2)$$
$$= \Big\{ \boldsymbol{v}(\boldsymbol{x}, t) = (W_L \sigma(\cdot) + \boldsymbol{b}_L) \circ (W_{L-1}\sigma(\cdot) + \boldsymbol{b}_{L-1}) \circ \cdots \circ$$
$$(W_1 \sigma(\cdot) + \boldsymbol{b}_1)([\boldsymbol{x}^T, t]^T) : \text{ network width bounded by } M,$$
$$\sup_{\boldsymbol{x}, t} \|\boldsymbol{v}(\boldsymbol{x}, t)\| \leq K, \ \max\{\|\boldsymbol{b}_i\|_\infty, \|W_i\|_\infty\} \leq \kappa$$
$$\text{for } i = 1, \cdots, L, \sum_{i=1}^{L}(\|W_i\|_0 + \|\boldsymbol{b}_i\|_0) \leq J,$$
$$\|\boldsymbol{v}(\boldsymbol{x}_1, t) - \boldsymbol{v}(\boldsymbol{x}_2, t)\| \leq \gamma_1 \|\boldsymbol{x}_1 - \boldsymbol{x}_2\| \text{ for any } t \in [0, T],$$
$$\|\boldsymbol{v}(\boldsymbol{x}, t_1) - \boldsymbol{v}(\boldsymbol{x}, t_2)\| \leq \gamma_2 \|t_1 - t_2\| \text{ for any } \boldsymbol{x} \Big\},$$

where the network width refers to the maximum dimensions of the weight matrices, $\sigma$ is the ReLU activation, and $\|\cdot\|_\infty$ and $\|\cdot\|_0$ denote the maximum magnitude of entries and the number of nonzero entries, respectively. In the sequel, we write the neural network class as NN for brevity.

**Theorem 3.1.** *Suppose Assumption 1.1 holds. Given an approximation error $\varepsilon > 0$, for any velocity field $\boldsymbol{v}^*$ with Lipschitz constant $\zeta$ w.r.t. $\boldsymbol{x}$ on $[0, T]$, we choose the hypoth-*

*esis class* NN *with*

$$L = \mathcal{O}\left(d + \log \frac{1}{\varepsilon}\right),$$

$$M = \mathcal{O}\left(\frac{d^{3/2}(\log(d/\varepsilon))^{\frac{d+1}{2}}}{(1-T)^4}\zeta^d \varepsilon^{-(d+1)}\right),$$

$$J = \mathcal{O}\left(\frac{d^{3/2}(\log(d/\varepsilon))^{\frac{d+1}{2}}}{(1-T)^4}\zeta^d \varepsilon^{-(d+1)}\left(\log \frac{1}{\varepsilon} + d\right)\right),$$

$$K = \mathcal{O}\left(\frac{\sqrt{d \log \frac{d}{\varepsilon}}}{1-T}\right),$$

$$\kappa = \mathcal{O}\left(\zeta\sqrt{\log(d/\varepsilon)} \vee \frac{\sqrt{d^3 \log(d/\varepsilon)}}{(1-T)^4}\right), \ \gamma_1 = 10d\zeta,$$

$$\gamma_2 = \mathcal{O}\left(\frac{\sqrt{d^3 \log(d/\varepsilon)}}{(1-T)^4}\right).$$

*There exists an $\widehat{\boldsymbol{v}}_\theta \in$ NN, such that for any $t \in [0, T]$, we have*

$$\|\widehat{\boldsymbol{v}}_\theta(\cdot, t) - \boldsymbol{v}^*(\cdot, t)\|_{L^2(\pi_t)} \leq (\sqrt{d} + 1)\varepsilon,$$

*where $\pi_t$ is the distribution of $X_t = tX_1 + (1-t)X_0$.*

The proof of Theorem 3.1 can be found in Appendix A.1.

**Universal Approximation under the $L^2$-norm** Many existing universal approximation theory of neural networks focus on approximating target functions on a compact domain under the $L^\infty$-norm (Yarotsky, 2017; Schmidt-Hieber, 2020; Gühring et al., 2020). Instead, we provide an $L^2$-approximation error bound over the unbounded input domain, where we tackle the unboundedness through a truncation argument.

**Lipschitz Neural Network** Conventional universal approximation theories of neural networks do not typically provide guarantees on the Lipschitz continuity of the network (Cybenko, 1989; Barron, 1993; Yarotsky, 2017), which is important for effective learning of the true velocity field. A line of research (Jiao et al., 2023; Dahal et al., 2022; Huang et al., 2022) studies Lipschitz neural networks motivated by the Wasserstein Generative Adversarial Network (WGAN) (Arjovsky et al., 2017). The paper (Jiao et al., 2023) studies the approximation capacity of ReLU neural networks with norm constraints on the weights. Meanwhile, (Huang et al., 2022, Lemma 11) provides an explicit bound on the Lipscitz constant required for approximating Hölder functions. In (Dahal et al., 2022), statistical guarantees for WGAN are provided under the Wasserstein 1-distance, assuming that the data distribution is supported on a low-dimensional manifold. These techniques are scalable to our analysis, and for brevity, we adopt the proof of the work (Chen et al., 2023a). The key difference between our paper and (Chen et al., 2023a) is that they assume the on-support score function is Lipschitz uniformly for $t \in [t_0, T]$, whereas our paper derives the Lipschitzness of the true velocity field from the assumption on the target distribution. In our construction, the Lipschitz continuity constraints $\gamma_1$ and $\gamma_2$ do not undermine the approximation power of the neural networks. In practice, such Lipschitz regularity is often enforced during training by adding regularization (Virmaux & Scaman, 2018; Pauli et al., 2021; Gouk et al., 2021). From a theoretical perspective, the Lipschitz property of the estimated velocity field is crucial in bounding the distribution recovery error, as we demonstrate in Section 5. Moreover, the Lipschitz continuity of the estimated velocity field ensures the existence and uniqueness of the solution of the ODE.

**Time as an Additional Input Dimension** In our approach, we introduce time $t$ as an extra input dimension to the neural network, and the network size scales polynomially with the Lipschitz constant $\tau$ of the true velocity field with respect to $t$. In Section D, we derive an upper bound for $\tau$ on a clipped time span $[0, T]$, where $T < 1$.

**Proof Sketch** Theorem 3.1 is established by construction. A noteworthy distinction from the existing universal approximation theories is that the input domain of the velocity field is unbounded. To establish the theorem, we leverage a truncation argument. Let $R$ be a truncation radius. On the hypercube $[-R, R]^d \times [0, T]$, we construct $\overline{\boldsymbol{v}}_\theta$ as a piece-wise linear function to approximate $\boldsymbol{v}^*$ in the sense of $L^\infty([-R, R]^d \times [0, T])$. Outside the hypercube, we simply set $\overline{\boldsymbol{v}}_\theta = 0$. The $L^2$ approximation error can be decomposed as

$$
\|\overline{\boldsymbol{v}}_\theta(\cdot, t) - \boldsymbol{v}^*(\cdot, t)\|_{L^2(\pi_t)}
$$

$$
= \underbrace{\left( \int_{\|\boldsymbol{x}\| \le R} \|\overline{\boldsymbol{v}}_\theta(\boldsymbol{x}, t) - \boldsymbol{v}^*(\boldsymbol{x}, t)\|^2 \pi_t(\mathrm{d}\boldsymbol{x}) \right)^{1/2}}_{(I)}
$$

$$
+ \underbrace{\left( \int_{\|\boldsymbol{x}\| > R} \|\overline{\boldsymbol{v}}_\theta(\boldsymbol{x}, t) - \boldsymbol{v}^*(\boldsymbol{x}, t)\|^2 \pi_t(\mathrm{d}\boldsymbol{x}) \right)^{1/2}}_{(II)}.
$$

The error term (I) is directly bounded by the approximation error of $\overline{\boldsymbol{v}}_\theta$ on the hypercube. It is worth noting that since $\overline{\boldsymbol{v}}_\theta$ is bounded and $\boldsymbol{v}^*(X_t, t)$ has a bounded second moment, the term (II) can be controlled by utilizing the tail behavior of $\pi_t$.

# 4. Generalization

In this section, we consider the generalization error of estimating the velocity field. We begin with the following connection between the loss function $\mathcal{L}(\boldsymbol{v})$ and the $L^2$ approximation error $\|\boldsymbol{v}(\cdot, t) - \boldsymbol{v}^*(\cdot, t)\|_{L^2(\pi_t)}$.

**Lemma 4.1.** *The following holds for any* $\boldsymbol{v}(\boldsymbol{x}, t)$:

$$
\mathcal{L}(\boldsymbol{v}) - \mathcal{L}(\boldsymbol{v}^*) = \frac{1}{T} \int_0^T \|\boldsymbol{v}(\cdot, t) - \boldsymbol{v}^*(\cdot, t)\|_{L^2(\pi_t)}^2 \mathrm{d}t.
$$

*Proof.* By some calculus, we have

$$
\mathbb{E}\left[ \|X_1 - X_0 - \boldsymbol{v}(X_t, t)\|^2 \right]
$$
$$
= \mathbb{E}\left[ \|X_1 - X_0 - \boldsymbol{v}^*(X_t, t) + \boldsymbol{v}^*(X_t, t) - \boldsymbol{v}(X_t, t)\|^2 \right]
$$
$$
= \mathbb{E}\left[ \|X_1 - X_0 - \boldsymbol{v}^*(X_t, t)\|^2 \right] + \|\boldsymbol{v}(\cdot, t) - \boldsymbol{v}^*(\cdot, t)\|_{L^2(\pi_t)}^2
$$
$$
+ 2\mathbb{E}\left[ \langle X_1 - X_0 - \boldsymbol{v}^*(X_t, t), \boldsymbol{v}^*(X_t, t) - \boldsymbol{v}(X_t, t) \rangle \right]. \tag{7}
$$

By taking expectation conditioned on $X_t$, we have

$$
\mathbb{E}\left[ \langle X_1 - X_0 - \boldsymbol{v}^*(X_t, t), \boldsymbol{v}^*(X_t, t) - \boldsymbol{v}(X_t, t) \rangle \right]
$$
$$
= \mathbb{E}\left[ \mathbb{E}[\langle X_1 - X_0 - \boldsymbol{v}^*(X_t, t), \boldsymbol{v}^*(X_t, t) - \boldsymbol{v}(X_t, t) \rangle | X_t] \right]
$$
$$
= \mathbb{E}\left[ \langle \mathbb{E}[X_1 - X_0 | X_t] - \boldsymbol{v}^*(X_t, t), \boldsymbol{v}^*(X_t, t) - \boldsymbol{v}(X_t, t) \rangle \right]
$$
$$
= \mathbb{E}\left[ \langle \boldsymbol{v}^*(X_t, t) - \boldsymbol{v}^*(X_t, t), \boldsymbol{v}^*(X_t, t) - \boldsymbol{v}(X_t, t) \rangle \right] = 0.
$$

Substituting the above identity into (7) and integrating on interval $[0, T]$, we obtain

$$
\mathcal{L}(\boldsymbol{v}) = \mathcal{L}(\boldsymbol{v}^*) + \frac{1}{T} \int_0^T \|\boldsymbol{v}(\cdot, t) - \boldsymbol{v}^*(\cdot, t)\|_{L^2(\pi_t)}^2 \mathrm{d}t,
$$

which concludes the proof. $\qquad \square$

According to Lemma 4.1, minimizing (4) is equivalent to minimizing the difference between the network and the true velocity field in $L^2(\pi_t)$-norm.

**Empirical Evaluation** Let us define

$$\ell(\boldsymbol{x}, \boldsymbol{v}) := \frac{1}{T} \int_0^T \int \|\boldsymbol{x} - \boldsymbol{x}_0 - \boldsymbol{v}(t\boldsymbol{x} + (1-t)\boldsymbol{x}_0, t)\|^2$$
$$\pi_0(\boldsymbol{x}_0)\mathrm{d}\boldsymbol{x}_0\mathrm{d}t. \tag{8}$$

In this paper, we choose the standard Gaussian distribution as the prior distribution, i.e., $\pi_0 = \mathcal{N}(0, I_d)$, where $d$ is the dimension of the data. Given $n$ independent and identically distributed (i.i.d.) samples $\{\boldsymbol{x}_{1,i}\}_{i=1}^n$ from $\pi_1$, we have the following empirical version of the least square loss:

$$\overline{\mathcal{L}}(\boldsymbol{v}) := \frac{1}{n} \sum_{i=1}^n \ell(\boldsymbol{x}_{1,i}, \boldsymbol{v}). \tag{9}$$

Since our main interest lies in the sample complexity of sampling from $\pi_1$, we consider the situation where $\ell(\boldsymbol{x}, \boldsymbol{v})$ can be computed exactly. However, in the usual implementation, the expectation in (8) is replaced by empirical evaluation. Given $m$ i.i.d. samples $\{(t_j, \boldsymbol{x}_{0,j})\}_{j=1}^m$ from $\mathrm{Unif}[0, T]$ and $\pi_0$, which are cheap to generate, then (8) has the following empirical evaluation:

$$\widehat{\ell}(\boldsymbol{x}, \boldsymbol{v}) := \frac{1}{m} \sum_{j=1}^m \|\boldsymbol{x} - \boldsymbol{x}_{0,j} - \boldsymbol{v}(t_j\boldsymbol{x} + (1-t_j)\boldsymbol{x}_{0,j}, t_j)\|^2. \tag{10}$$

Due to the efficacy of sampling $t$ and $\boldsymbol{x}_0$, $\ell(\boldsymbol{x}, \boldsymbol{v})$ can be efficiently approximated by $\widehat{\ell}(\boldsymbol{x}, \boldsymbol{v})$ via polynomial-size sample from $\mathrm{Unif}[0, T]$ and $\pi_0$, which will be explained exactly in Section 4.1. Now, we consider the Empirical Risk Minimization (ERM):

$$\widehat{\boldsymbol{v}} \in \operatorname*{argmin}_{\boldsymbol{v} \in \mathcal{V}} \left\{ \widehat{\mathcal{L}}(\boldsymbol{v}) := \frac{1}{n} \sum_{i=1}^n \widehat{\ell}(\boldsymbol{x}_{1,i}, \boldsymbol{v}) \right\} \tag{11}$$

### 4.1. Error Decomposition

The error of the estimated vector field (11) can be decomposed as:

$$\mathcal{L}(\widehat{\boldsymbol{v}}) - \mathcal{L}(\boldsymbol{v}^*) = \underbrace{\mathcal{L}(\widehat{\boldsymbol{v}}) - \inf_{\boldsymbol{v} \in \mathrm{NN}} \mathcal{L}(\boldsymbol{v})}_{\text{Generalization error}}$$
$$+ \underbrace{\inf_{\boldsymbol{v} \in \mathrm{NN}} (\mathcal{L}(\boldsymbol{v}) - \mathcal{L}(\boldsymbol{v}^*))}_{\text{Approximation error}} \tag{12}$$

Further, the generalization error has the following decomposition:

$$\mathcal{L}(\widehat{\boldsymbol{v}}) - \inf_{\boldsymbol{v} \in \mathrm{NN}} \mathcal{L}(\boldsymbol{v}) = \mathcal{L}(\widehat{\boldsymbol{v}}) - \widehat{\mathcal{L}}(\widehat{\boldsymbol{v}}) + \widehat{\mathcal{L}}(\widehat{\boldsymbol{v}}) - \widehat{\mathcal{L}}(\widetilde{\boldsymbol{v}})$$
$$+ \widehat{\mathcal{L}}(\widetilde{\boldsymbol{v}}) - \mathcal{L}(\widetilde{\boldsymbol{v}})$$
$$\leq \mathcal{L}(\widehat{\boldsymbol{v}}) - \widehat{\mathcal{L}}(\widehat{\boldsymbol{v}}) + \widehat{\mathcal{L}}(\widetilde{\boldsymbol{v}}) - \mathcal{L}(\widetilde{\boldsymbol{v}}), \tag{13}$$

where the inequality follows from ERM, and $\widetilde{v}$ is defined in (6). Note that, for any $\boldsymbol{v} \in \mathrm{NN}$, we have

$$\mathcal{L}(\boldsymbol{v}) - \widehat{\mathcal{L}}(\boldsymbol{v}) = \mathcal{L}(\boldsymbol{v}) - \overline{L}(\boldsymbol{v}) + \overline{L}(\boldsymbol{v}) - \widehat{\mathcal{L}}(\boldsymbol{v})$$
$$= \frac{1}{n} \sum_{i=1}^n (\mathcal{L}(\boldsymbol{v}) - \ell(\boldsymbol{x}_{1,i}, \boldsymbol{v})) \tag{14}$$
$$+ \frac{1}{n} \sum_{i=1}^n (\ell(\boldsymbol{x}_{1,i}, \boldsymbol{v}) - \widehat{\ell}(\boldsymbol{x}_{1,i}, \boldsymbol{v}))$$

By defining $\mathcal{H} = \{\ell(\cdot, \boldsymbol{v}) : \boldsymbol{v} \in \mathrm{NN}(L, M, J, K, \kappa, \gamma_1, \gamma_2)\}$, we can apply conventional statistical learning arguments to analyze the first term within the function class $\mathcal{H}$. Due to the unbounded nature of the loss function $|\boldsymbol{x} - \boldsymbol{x}_0 - \boldsymbol{v}(t\boldsymbol{x} + (1-t)\boldsymbol{x}_0, t)|^2$, controlling the second term requires an additional truncation argument. We will provide further details on our approach at the end of this section.

The complexity of a function class can be measured using the covering number.

**Definition 4.2** (Covering number). Let $\rho$ be a pseudo-metric on $\mathcal{M}$ and $S \subseteq \mathcal{M}$. For any $\delta > 0$, a set $A \subseteq \mathcal{M}$ is called a $\delta$-covering of $S$ if for any $x \in S$ there exists $y \in A$ such that $\rho(x, y) \leq \delta$. The $\delta$-covering number of $S$, denoted by $\mathcal{N}(\delta, S, \rho)$, is the minimum cardinality of any $\delta$-covering of $S$.

The function class $\mathcal{H}$ exhibits the following properties, which are useful for analyzing the generalization error.

**(i) Bounded sup-norm** According to Theorem 3.1, the estimated velocity field $\widehat{\boldsymbol{v}}(\boldsymbol{x}, t)$ can be chosen to satisfy the condition $\|\widehat{\boldsymbol{v}}\|_{L^\infty(\mathbb{R}^d \times [0,T])} \leq K = \mathcal{O}\left(\frac{\sqrt{\log(d/\varepsilon)}}{1-T}\right)$. Then Lemma B.1 shows that

$$\sup_{\boldsymbol{v} \in \mathrm{NN}} \sup_{\boldsymbol{x} \in [0,1]^d} \ell(\boldsymbol{x}, \boldsymbol{v}) \lesssim d + K^2 \lesssim d + \frac{\log(d/\varepsilon)}{(1-T)^2}.$$

**(ii) Covering number evaluation** The covering number of the network class selected in Theorem 3.1 is evaluated as follows:

$$\log \mathcal{N}(\delta, \mathrm{NN}, \|\cdot\|_{L^\infty([-D,D]^d \times [0,1])})$$
$$\lesssim JL \log\left(\frac{LM(D \vee 1)\kappa}{\delta}\right). \tag{15}$$

The above evaluation can be found in (Chen et al., 2022b, Lemma 5.3). Based on the above result, we have the following evaluation for the covering number of the loss function class $\mathcal{H}$:

**Lemma 4.3.** *The covering number of $\mathcal{H}$ is evaluated by*

$$\log \mathcal{N}(\delta, \mathcal{H}, \|\cdot\|_{L^\infty([0,1]^d)})$$
$$\lesssim JL \log \left( \frac{(K + d^{1/2})LM\kappa\sqrt{\log((K^2 + d)/\delta)}}{\delta} \right).$$
(16)

The proof of Lemma 4.3 is deferred to Appendix B.2. It is worth noting that the evaluation is non-trivial because the evaluation in (15) considers the $L^\infty$-norm on a bounded subspace, while the region of integration in (8) is unbounded. To overcome this challenge, we utilize a truncation argument to provide the covering number evaluation for $\mathcal{H}$.

Based on the above discussion, we can now derive the following generalization bound

**Theorem 4.4.** *Suppose Assumption 1.1 holds. For any velocity field $\boldsymbol{v}^*$ with Lipschitz constant $\zeta$ w.r.t. $\boldsymbol{x}$, given $n$ samples $\{\boldsymbol{x}_{1,i}\}_{i=1}^n$ from $\pi_1$ and $m$ samples from $\pi_0$ and $\mathrm{Unif}[0,T]$, we choose NN as in Theorem 3.1 with $\varepsilon = n^{-\frac{1}{d+5}}$. Then with probability of at least $1 - \frac{1}{n}$, it holds*

$$\frac{1}{T} \int_0^T \|\widehat{\boldsymbol{v}}(\cdot, t) - \boldsymbol{v}^*(\cdot, t)\|_{L^2(\pi_t)}^2 \mathrm{d}t$$
$$= \widetilde{\mathcal{O}} \left( \frac{\zeta^{d/2}}{(1-T)^4} \left( n^{-\frac{2}{d+5}} + n^{\frac{d+1}{2(d+5)}} m^{-\frac{1}{2}} \right) \right),$$

*where we omit factors in $d, \log n, \log m, \log(1 - T)$. By setting $m$ to be of the order $\mathcal{O}(n)$, we obtain the convergence rate of order $\widetilde{\mathcal{O}} \left( \frac{\zeta^{d/2}}{(1-T)^4} n^{-\frac{2}{d+5}} \right)$.*

The proof can be found in Appendix B.3. To the best of our knowledge, Theorem 4.4 provides the first explicit sample complexity bound for FM. Theorem 4.4 becomes vacuous when $T$ tends to 1 with fixed sample size $n$. This is a consequence of the blowup of the velocity field $\boldsymbol{v}^*(\boldsymbol{x}, t)$ as $t$ tends to 1. Although a smaller early stopping time leads to better generalization error, stopping the sampling process at an early time results in a bad distribution recovery. In Section 5, we will show the tradeoff in the choice of stopping time $T$.

**Proof Sketch** The generalization error is divided into two terms. The first term's randomness arises from drawing samples from the target distribution $\pi_1$, while the second term's randomness comes from sampling from $\pi_0$ and $\mathrm{Unif}[0, T]$. We encounter two difficulties in deriving the generalization error bound. The first difficulty lies in evaluating the covering number of the loss function class $\mathcal{H}$ for the first term. The second difficulty stems from the unboundedness of the term $\|\boldsymbol{x} - \boldsymbol{x}_0 - \boldsymbol{v}(t\boldsymbol{x} + (1-t)\boldsymbol{x}_0, t)\|^2$ in the second term. To handle this, we leverage the concentration property of the Gaussian prior distribution and employ a truncation argument to provide an upper bound for the second term

with high probability. Specifically, the second term can be decomposed as follows:

$$\ell(\boldsymbol{x}_{1,i}, \boldsymbol{v}) - \widehat{\ell}(\boldsymbol{x}_{1,i}, \boldsymbol{v}) = \underbrace{\ell(\boldsymbol{x}_{1,i}, \boldsymbol{v}) - \ell^{\mathrm{trunc}}(\boldsymbol{x}_{1,i}, \boldsymbol{v})}_{\text{Truncation error (I)}}$$
$$+ \underbrace{\ell^{\mathrm{trunc}}(\boldsymbol{x}_{1,i}, \boldsymbol{v}) - \widehat{\ell}^{\mathrm{trunc}}(\boldsymbol{x}_{1,i}, \boldsymbol{v})}_{\text{Statistical error}}$$
$$+ \underbrace{\widehat{\ell}^{\mathrm{trunc}}(\boldsymbol{x}_{1,i}, \boldsymbol{v}) - \widehat{\ell}(\boldsymbol{x}_{1,i}, \boldsymbol{v})}_{\text{Truncation error (II)}},$$

where $\ell^{\mathrm{trunc}}(\boldsymbol{x}_{1,i}, \boldsymbol{v}) := \mathbb{E}_{t,\boldsymbol{x}_0}[\|\boldsymbol{x}_{1,i} - \boldsymbol{x}_0 - \boldsymbol{v}(t\boldsymbol{x} + (1 - t)\boldsymbol{x}_0, t)\|^2 \mathbb{1}\{\|\boldsymbol{x}_0\|_\infty \leq R\}]$ and $\widehat{\ell}^{\mathrm{trunc}}(\boldsymbol{x}_{1,i}, \boldsymbol{v}) := \frac{1}{m} \sum_{j=1}^m \|\boldsymbol{x}_{1,i} - \boldsymbol{x}_{0,j} - \boldsymbol{v}(t_j \boldsymbol{x}_{1,i} + (1 - t)\boldsymbol{x}_{0,j}, t_j)\|^2 \mathbb{1}\{\|\boldsymbol{x}_{0,j}\|_\infty \leq R\}$. We can control Truncation error (I) by utilizing the concentration of Gaussian variables. On the other hand, Statistical error can be controlled using a covering number argument. Furthermore, Truncation error (II) is likely to be equal to zero due to the concentration of Gaussian variables.

# 5. Sampling

This section establishes distribution recovery guarantees using the estimated velocity field.

**Estimated Sampling Dynamics** Given the estimated velocity field $\widehat{\boldsymbol{v}}$, we can generate samples from an approximation of the continuous flow ODE starting from the prior distribution:

$$\mathrm{d}\widehat{X}_t(\boldsymbol{x}) = \widehat{\boldsymbol{v}}(\widehat{X}_t(\boldsymbol{x}), t)\mathrm{d}t, \ \widehat{X}_0(\boldsymbol{x}) = \boldsymbol{x} \sim \pi_0, \ 0 \leq t \leq T.$$
(17)

**Proposition 5.1.** *Suppose Assumption 1.1 holds. For any velocity field $\boldsymbol{v}^*$ with Lipschitz constant $\zeta$ w.r.t. $\boldsymbol{x}$, given $n$ samples $\{\boldsymbol{x}_{1,i}\}_{i=1}^n$ from $\pi_1$ and $m$ samples from $\pi_0$ and $\mathrm{Unif}[0,T]$, we choose NN as in Theorem 3.1 with $\varepsilon = n^{-\frac{1}{d+5}}$. Then with probability of at least $1 - \frac{1}{n}$, it holds*

$$W_2(\pi_T, \widehat{\pi}_T) = \widetilde{\mathcal{O}} \left( e^{\gamma_1} \frac{\zeta^{d/4}}{(1-T)^2} n^{-\frac{1}{d+5}} \right).$$
(18)

*Proof.* Note that $X_t(\boldsymbol{x})$ and $\widehat{X}_t(\boldsymbol{x})$ form a coupling of $\pi_t$ and $\widehat{\pi}_t$, by the definition of Wasserstein-2 distance, we have

$$W_2^2(\pi_t, \widehat{\pi}_t) \leq \int_{R^d} \|X_t(\boldsymbol{x}) - \widehat{X}_t(\boldsymbol{x})\|^2 \pi_0(\boldsymbol{x})\mathrm{d}\boldsymbol{x}, \quad (19)$$

where $X_t$ is the flow map solution of (53) with the exact $\boldsymbol{v}^*$ defined in (3) and $\widehat{X}_t$ is the flow map solution of (54). Now, we consider the evolution of

$$R_t := \int_{\mathbb{R}^d} \|X_t(\boldsymbol{x}) - \widehat{X}_t(\boldsymbol{x})\|^2 \pi_0(\boldsymbol{x})\mathrm{d}\boldsymbol{x}.$$

Differentiating on both sides, we get

$$
\begin{aligned}
\frac{\mathrm{d}R_t}{\mathrm{d}t} = & 2\int_{\mathbb{R}^d}\langle \boldsymbol{v}^*(X_t(\boldsymbol{x}),t) - \widehat{\boldsymbol{v}}(\widehat{X}_t(\boldsymbol{x}),t), X_t(\boldsymbol{x}) - \widehat{X}_t(\boldsymbol{x})\rangle \\
& \pi_0(\boldsymbol{x})\mathrm{d}\boldsymbol{x}
\end{aligned}
$$

$$
\begin{aligned}
= & 2\int_{\mathbb{R}^d}\langle \boldsymbol{v}^*(X_t(\boldsymbol{x}),t) - \widehat{\boldsymbol{v}}(X_t(\boldsymbol{x}),t) + \widehat{\boldsymbol{v}}(X_t(\boldsymbol{x}),t) \\
& -\widehat{\boldsymbol{v}}(\widehat{X}_t(\boldsymbol{x}),t), X_t(\boldsymbol{x}) - \widehat{X}_t(\boldsymbol{x})\rangle\pi_0(\boldsymbol{x})\mathrm{d}\boldsymbol{x}.
\end{aligned}
\tag{20}
$$

Using the inequality $2\langle a, b\rangle \leq \|a\|^2 + \|b\|^2$, we have

$$
\begin{aligned}
& 2\langle \boldsymbol{v}^*(X_t(\boldsymbol{x}),t) - \widehat{\boldsymbol{v}}(X_t(\boldsymbol{x}),t), X_t(\boldsymbol{x}) - \widehat{X}_t(\boldsymbol{x})\rangle \leq \\
& \|\boldsymbol{v}^*(X_t(\boldsymbol{x}),t) - \widehat{\boldsymbol{v}}(X_t(\boldsymbol{x}),t)\|^2 + \|X_t(\boldsymbol{x}) - \widehat{X}_t(\boldsymbol{x})\|^2.
\end{aligned}
\tag{21}
$$

Note that $\widehat{\boldsymbol{v}} \in \mathrm{NN}$ defined in Theorem 3.1 is $\gamma_1$-Lipschitz continuous w.r.t. $\boldsymbol{x}$, the Cauchy-Schwartz inequality implies

$$
\begin{aligned}
& 2\langle \widehat{\boldsymbol{v}}(X_t(\boldsymbol{x}),t) - \widehat{\boldsymbol{v}}(\widehat{X}_t(\boldsymbol{x}),t), X_t(\boldsymbol{x}) - \widehat{X}_t(\boldsymbol{x})\rangle \\
& \leq 2\gamma_1\|X_t(\boldsymbol{x}) - \widehat{X}_t(\boldsymbol{x})\|^2.
\end{aligned}
\tag{22}
$$

Combining (20), (21) and (22), we obtain

$$
\begin{aligned}
\frac{\mathrm{d}R_t}{\mathrm{d}t} \leq & (1 + 2\gamma_1)R_t \\
& + \int_{\mathbb{R}^d}\|\boldsymbol{v}^*(X_t(\boldsymbol{x}),t) - \widehat{\boldsymbol{v}}(X_t(\boldsymbol{x}),t)\|^2\pi_0(\boldsymbol{x})\mathrm{d}\boldsymbol{x}.
\end{aligned}
$$

Therefore, by Lemma C.6 and since $R_0 = 0$, we deduce

$$
\begin{aligned}
R_T \leq & e^{1+2\gamma_1}\int_0^T\int_{\mathbb{R}^d}\|\boldsymbol{v}^*(X_t(\boldsymbol{x}),t) - \widehat{\boldsymbol{v}}(X_t(\boldsymbol{x}),t)\|^2 \\
& \pi_0(\boldsymbol{x})\mathrm{d}\boldsymbol{x}\mathrm{d}t \\
= & e^{1+2\gamma_1}\int_0^T\|\boldsymbol{v}^*(\cdot,t) - \widehat{\boldsymbol{v}}(\cdot,t)\|_{L^2(\pi_t)}^2\mathrm{d}t.
\end{aligned}
$$

By Theorem 4.4 and the fact that $\widehat{\boldsymbol{v}}$ is $\gamma_1$-Lipschitz continuous w.r.t. $\boldsymbol{x}$ since we choose NN as in Theorem 4.4, we get the desired result. □

**Time Discretization** In practice, we need to use a discrete-time approximation for the sampling dynamics (17). Let $0 = t_0 < t_1 < \cdots < t_N = T$ be the discretization points. We consider the explicit Euler discretization scheme:

$$
\mathrm{d}\widetilde{X}_t(\boldsymbol{x}) = \widehat{\boldsymbol{v}}(\widetilde{X}_{t_k}(\boldsymbol{x}), t_k)\mathrm{d}t, \ t \in [t_k, t_{k+1}),
\tag{23}
$$

for $k = 0, 1, \ldots, N-1$ and $\widetilde{X}_0(\boldsymbol{x}) = \boldsymbol{x} \sim \pi_0$. We denote the distribution of $\widetilde{X}_T(\boldsymbol{x})$ by $\widetilde{\pi}_T$.

To establish the distribution recovery guarantees, we need the following discretization error bound:

**Lemma 5.2.** *Let $0 = t_0 < t_1 < \cdots < t_N = T$ be the discretization points. For any neural network $\widehat{\boldsymbol{v}}$ in*

$\mathrm{NN}(L, M, J, K, \kappa, \gamma_1, \gamma_2)$, *we have:*

$$
W_2(\widehat{\pi}_T, \widetilde{\pi}_T) = \mathcal{O}\left(e^{\gamma_1}(\gamma_1 K + \gamma_2)\sqrt{\sum_{k=0}^{N-1}(t_{k+1} - t_k)^3}\right),
$$

*where $\widehat{\pi}$ is the distribution of the final output of the estimated sampling dynamics (17).*

The proof of Lemma 5.2 can be found in Appendix C.2.

**Tradeoff on Stopping Time** $T$ To show the tradeoff, we first present the following lemma:

**Lemma 5.3.** *Suppose Assumption 1.1 holds, we have*

$$
W_2(\pi_T, \pi_1) \lesssim (1 - T)\sqrt{d}.
$$

The proof of Lemma 5.3 is deferred to Appendix C.2. Proposition 5.1 demonstrates that as the stopping time $T$ tends to 1, the error of using the estimated velocity field in the sampling dynamics increases. Conversely, according to Lemma 5.3, the Wasserstein-2 distance between $\pi_T$ and $\pi_1$ decreases as $T$ approaches 1. This reveals a tradeoff in the stopping time $T$ between the error in velocity field estimation and the distribution recovery.

# 6. Conclusion

This paper presents a statistical learning theory perspective on CNFs based on FM. We demonstrate that a Lipschitz neural network can approximate the true velocity field under $L^2(\pi_t)$-norm and provide a sample complexity analysis for estimating the velocity field. Furthermore, we prove that under mild assumptions, the generated distribution of CNFs based on FM converges to the target data in Wasserstein-2 distance. Additionally, we show that the convergence rate can be significantly improved by assuming an additional mild Lipschitz condition on the target score function. To the best of our knowledge, this is the first end-to-end analysis of FM.

# Acknowledgement

This work is supported by the Key R&D Program of Hubei Province under Grant 2024BAB038, the National Key R&D Program of China under Grant 2023YFC3604702, the Fundamental Research Funds for the Central Universities under Grant 2042025kf0045.

# Impact Statement

This paper presents work whose goal is to advance the field of Machine Learning. There are many potential societal consequences of our work, none of which we feel must be specifically highlighted here.

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

# A. Approximation Error

## A.1. Proof of Theorem 3.1

*Proof.* The goal is to find a network $\widehat{v}$ in NN to approximate the true vector field $v^*$. A major difficulty in approximating $v^*(x, t)$ is that the input space $\mathbb{R}^d \times [0, T]$ is unbounded. To address this difficulty, we partition $\mathbb{R}^d$ into a compact subset $\mathcal{K}$ and its complement $\mathcal{K}^c$. On $\mathcal{K} \times [0, T]$, we construct $\widehat{v}$ to achieve an $L^\infty$ approximation. On the $\mathcal{K}^c$, we simply set $\widehat{v}(x, t) = 0$. Since we assume $\pi_1$ is supported on a compact set, the $L^2(\pi_t)$ approximation error of $\widehat{v}(x, t)$ to $v^*(x, t)$ can still be controlled.

• **Approximation on $\mathcal{K} \times [0, T]$.** We choose $\mathcal{K} = \{x \| \|x\|_\infty \le R\}$ to be a $d$-dimensional hypercube with edge length $2R > 0$, where $R$ will be determined later. On $\mathcal{K} \times [0, T]$, we approximate $k$-coordinate maps $v_k^*(x, t)$ separately, where $v^* = [v_1^*(x, t), \cdots, v_d^*(x, t)]^T$.

First, we rescale the input by $x' = \frac{1}{2R}(x + R\mathbf{1})$ and $t' = t/T$, where $\mathbf{1} := [1, \cdots, 1]^T$, so that the transformed space is $[0, 1]^d \times [0, 1]$. Such a transformation can be exactly implemented by a single ReLU layer.

By Lemma 1.4, $v^*(x, t)$ is $\zeta$-Lipschitz in $x$. We define the rescaled function on the transformed input space as $v(x', t') := v^*(2Rx' - R\mathbf{1}, Tt')$, so that $v$ is $2\zeta R$-Lipschitz in $x'$.

We also denote the Lipschitz constant of $v(x', t')$ w.r.t. $t'$ as $T\tau(R)$, when $x' \in [0, 1]^d$. We denote

$$\tau(R) := \sup_{t \in [0, T]} \sup_{x \in [-R, R]^d} \|\partial_t v^*(x, t)\|$$

An upper bound for $\tau(R)$ is computed in Lemma D.4 by $\tau(R) = \mathcal{O}\left(\frac{d^{3/2}(R+1)}{(1-T)^4}\right)$. Now the goal becomes approximating $v$ on $[0, 1]^d \times [0, 1]$.

Second, we partition $[0, 1]^d$ into non-overlapping hypercubes with equal edge length $e_1$. We also partition the time interval $[0, 1]$ into non-overlapping sub-intervals of length $e_2$. $e_1$ and $e_2$ will be chosen depending on the desired approximation error. We denote $N_1 = \lceil \frac{1}{e_1} \rceil$ and $N_2 = \lceil \frac{1}{e_2} \rceil$.

Let $\boldsymbol{m} = [m_1, \cdots, m_d]^T \in [N_1]^d$ be a multi-index. We define $\overline{v}$ as

$$\overline{v}_i(x', t') := \sum_{\boldsymbol{m} \in [N_1]^d, j \in [N_2]} v_i^*\left(2R\frac{\boldsymbol{m}}{N_1} - R\mathbf{1}, T\frac{j}{N_2}\right) \Psi_{\boldsymbol{m}, j}(x', t'),$$

where $\Psi_{\boldsymbol{m}, j}(x', t')$ is a partition of unity function, that is $\sum_{\boldsymbol{m} \in [N_1]^d, j \in [N_2]} \Psi_{\boldsymbol{m}, j}(x', t') \equiv 1$ on $[0, 1]^d \times [0, 1]$. We choose $\Psi_{\boldsymbol{m}, j}$ as a product of coordinate-wise trapezoid functions:

$$\Psi_{\boldsymbol{m}, j}(x', t') := \psi\left(3N_2\left(t' - \frac{j}{N_2}\right)\right) \prod_{i=1}^d \psi\left(3N_1\left(x_i' - \frac{m_i}{N_1}\right)\right)$$

where $\psi$ is a trapezoid function,

$$\psi(a) := \begin{cases} 1, & |a| < 1 \\ 2 - |a|, & |a| \in [1, 2] \\ 0, & |a| > 2. \end{cases}$$

We claim that

1. $\overline{v}_i$ is an approximation of $v_i$;

2. $\overline{v}_i$ can be implemented by a ReLU neural network $\widehat{v}_i$ with small error.

Both claims are verified in (Chen et al., 2020b, Lemma 10), where we only need to substitute the Lipschitz constant $2\zeta R$ and $\tau(R)$ into the error analysis. We use the coordinate-wise analysis in the proof of (Chen et al., 2020b, Lemma 10) for deriving the Lipschitz continuity w.r.t. $x'$ and $t'$. Similar proofs can be found in Huang et al. (2022). By concatenating $\widehat{v}_i$'s together, we construct $\widehat{v}_\theta = [\widehat{v}_1, \ldots, \widehat{v}_d]^T$. Given $\varepsilon$, if we achieve

$$\sup_{x', t' \in [0, 1]^d \times [0, 1]} \|\widehat{v}_\theta(x', t') - v(x', t')\|_\infty \le \varepsilon,$$

the neural network configuration is

$$L = \mathcal{O}\left(\log\frac{1}{\varepsilon} + d\right), \ M = \mathcal{O}\left(\tau(R)(\zeta R)^d \varepsilon^{-(d+1)}\right), \ J = \mathcal{O}\left(\tau(R)(\zeta R)^d \varepsilon^{-(d+1)}\left(\log\frac{1}{\varepsilon} + d\right)\right),$$

$$K = \mathcal{O}\left(\frac{\sqrt{d}R}{1-T}\right), \ \kappa = \max\left\{1, \zeta R, \tau(R)\right\}.$$

Here we already take $e_1 = \mathcal{O}\left(\frac{\varepsilon}{\zeta R}\right)$ and $e_2 = \mathcal{O}\left(\frac{\varepsilon}{\tau(R)}\right)$. The output range $K$ is computed by $K = \sqrt{d}\max_i \sup_{(\boldsymbol{x},t)\in[-R,R]^d\times[0,T]} \|v_i^*(\boldsymbol{x},t)\|$. Combining with the input transformation layer, i.e., $\boldsymbol{x} \to \boldsymbol{x}'$ and $t \to t'$ rescaling, we have constructed network is Lipschitz continuous in $\boldsymbol{x}'$, i.e., for any $\boldsymbol{x}_1, \boldsymbol{x}_2 \in \mathcal{K}$ and $t \in [0,T]$, it holds

$$\|\widehat{\boldsymbol{v}}_\theta(\boldsymbol{x}_1, t) - \widehat{\boldsymbol{v}}_\theta(\boldsymbol{x}_2, t)\|_\infty \leq 10d\zeta\|\boldsymbol{x}_1 - \boldsymbol{x}_2\|.$$

Moreover, the network is also Lipschitz in $t$, i.e., for any $t_1, t_2 \in [1, T]$ and $\|\boldsymbol{x}\|_\infty \leq R$, it holds

$$\|\widehat{\boldsymbol{v}}_\theta(\boldsymbol{x}, t_1) - \widehat{\boldsymbol{v}}_\theta(\boldsymbol{x}, t_2)\|_\infty \leq 10\tau(R)|t_1 - t_2|.$$

Due to the partition of unity function, $\Psi_{\boldsymbol{m},j}$ vanishes outside $\mathcal{K}$, we have $\widehat{\boldsymbol{v}}_\theta(\boldsymbol{x}, t) = 0$ for $\|\boldsymbol{x}\|_\infty > R$. Therefore the above Lipschitz continuity in $\boldsymbol{x}$ extends to $\mathbb{R}^d$.

• **Bounding $L^2$ approximation error.** The $L^2$ approximation error of $\widehat{\boldsymbol{v}}_\theta$ can be decomposed into two terms,

$$\|\boldsymbol{v}^*(\boldsymbol{x},t) - \widehat{\boldsymbol{v}}_\theta(\boldsymbol{x},t)\|_{L^2(\pi_t)} = \|(\boldsymbol{v}^*(\boldsymbol{x},t) - \widehat{\boldsymbol{v}}_\theta(\boldsymbol{x},t))\mathbb{1}\{\|\boldsymbol{x}\|_\infty \leq R\}\|_{L^2(\pi_t)}$$
$$+ \|(\boldsymbol{v}^*(\boldsymbol{x},t) - \widehat{\boldsymbol{v}}_\theta(\boldsymbol{x},t))\mathbb{1}\{\|\boldsymbol{x}\|_\infty > R\}\|_{L^2(\pi_t)}.$$

The first term on the right-hand side of the last display is bounded by

$$\|(\boldsymbol{v}^*(\boldsymbol{x},t) - \widehat{\boldsymbol{v}}_\theta(\boldsymbol{x},t))\mathbb{1}\{\|\boldsymbol{x}\|_\infty \leq R\}\|_{L^2(\pi_t)} \leq \sqrt{d}\sup_{(\boldsymbol{x},t)\in\mathcal{K}\times[0,T]} \|\boldsymbol{v}^*(\boldsymbol{x},t) - \widehat{\boldsymbol{v}}_\theta(\boldsymbol{x},t)\|_\infty \leq \sqrt{d}\varepsilon.$$

The second term admits an upper bound in Lemma A.1. Specifically, when choosing $R = \mathcal{O}\left(\sqrt{\log\frac{d}{\varepsilon}}\right)$, we have

$$\|(\boldsymbol{v}^*(\boldsymbol{x},t) - \widehat{\boldsymbol{v}}_\theta(\boldsymbol{x},t))\mathbb{1}\{\|\boldsymbol{x}\|_\infty > R\}\|_{L^2(\pi_t)} \leq \varepsilon.$$

As a result, with the choice of $R$, we obtain

$$\|\boldsymbol{v}^*(\boldsymbol{x},t) - \widehat{\boldsymbol{v}}_\theta(\boldsymbol{x},t)\|_{L^2(\pi_t)} \leq (\sqrt{d}+1)\varepsilon.$$

Substituting $R$ into the network configuration, we obtain

$$L = \mathcal{O}\left(d + \log\frac{1}{\varepsilon}\right), \ M = \mathcal{O}\left(\frac{d^{3/2}(\log(d/\varepsilon))^{\frac{d+1}{2}}}{(1-T)^4}\zeta^d\varepsilon^{-(d+1)}\right),$$

$$J = \mathcal{O}\left(\frac{d^{3/2}(\log(d/\varepsilon))^{\frac{d+1}{2}}}{(1-T)^4}\zeta^d\varepsilon^{-(d+1)}\left(\log\frac{1}{\varepsilon} + d\right)\right), \ K = \mathcal{O}\left(\frac{\sqrt{d\log\frac{d}{\varepsilon}}}{1-T}\right),$$

$$\kappa = \mathcal{O}\left(\zeta\sqrt{\log(d/\varepsilon)} \vee \frac{\sqrt{d^3\log(d/\varepsilon)}}{(1-T)^4}\right), \ \gamma_1 = 10d\zeta, \ \gamma_2 = \mathcal{O}\left(\frac{\sqrt{d^3\log(d/\varepsilon)}}{(1-T)^4}\right).$$

$\square$

## A.2. Truncation error

**Lemma A.1.** *Under Assumption 1.1, given $\varepsilon > 0$, with $R = \mathcal{O}\left(\sqrt{\log\frac{d}{\varepsilon}}\right)$, it holds*

$$\|(\boldsymbol{v}^*(\boldsymbol{x},t) - \widehat{\boldsymbol{v}}_\theta(\boldsymbol{x},t))\mathbb{1}\{\|\boldsymbol{x}\|_\infty > R\}\|_{L^2(\pi_t)} \leq \varepsilon.$$

*Proof.* For any $R > 0$, using the identity $\boldsymbol{v}^*(\boldsymbol{x}, t) = \mathbb{E}[X_1 - X_0 | X_t = \boldsymbol{x}]$, we have

$$
\begin{aligned}
&\int_{\{\|x\|_\infty\}>R} \|\boldsymbol{v}^*(x,t)\|^2 \pi_t(d\boldsymbol{x}) \\
&= \int_{\{\|x\|_\infty\}>R} \|\mathbb{E}[X_1 - X_0|X_t = \boldsymbol{x}]\|^2 \pi_t(d\boldsymbol{x}) \\
&\leq \int_{\{\|x\|_\infty\}>R} \mathbb{E}[\|X_1 - X_0\|^2 | X_t = \boldsymbol{x}]\pi_t(d\boldsymbol{x}) \\
&= \mathbb{E}_{X_t}\left[\mathbb{E}[\|X_1 - X_0\|^2 | X_t]\mathbb{1}\{\|X_t\|_\infty > R\}\right] \\
&\leq \mathbb{E}[\|X_1 - X_0\|^2 \mathbb{1}\{\|X_t\|_\infty > R\}] \\
&\leq \mathbb{E}[\|X_1 - X_0\|^4]^{1/2}\mathbb{P}(\|X_t\|_\infty > R)^{1/2},
\end{aligned}
\tag{24}
$$

where the second equality follows from the total expectation formula, and the last inequality follows from Cauchy-Schwartz inequality. Using the inequality $(a + b)^2 \leq 2a^2 + 2b^2$, we have the following upper bound for the fourth moment,

$$
\begin{aligned}
\mathbb{E}[\|X_1 - X_0\|^4] &\leq \mathbb{E}[(2\|X_1\|^2 + 2\|X_0\|^2)^2] \\
&\leq \mathbb{E}[4\|X_1\|^4 + 4\|X_0\|^4] \\
&\leq 4d^2 + 4\mathbb{E}\left[\left(\sum_{i=1}^d X_{0,i}^2\right)^2\right] \\
&= 4d^2 + 4\mathbb{E}\left[\sum_{k=1}^d X_{0,k}^4 + \sum_{i \neq j} X_{0,i}^2 X_{0,j}^2\right] \\
&= 8d(d + 1),
\end{aligned}
\tag{25}
$$

where $X_{0,i}$ denotes the $i$-coordinate of $X_0$. It remains to control the tail probability of $X_t$. Using the union inequality, we have

$$
\begin{aligned}
\mathbb{P}(\|X_t\|_\infty > R) &= \mathbb{P}\left(\bigcup_{i=1}^d \{|X_{t,i}| > R\}\right) \\
&\leq \sum_{i=1}^d \mathbb{P}(|X_{t,i}| > R).
\end{aligned}
$$

Thus, it suffices to control the tail probability of $X_{t,i}$ for $i = 1, \ldots, d$, where $X_{t,i}$ is the $i$-coordinate of $X_t$. Since we assume $\pi_1$ is supported on $[0, 1]^d$, we have

$$
\begin{aligned}
\mathbb{P}(|X_{t,i}| > R) &\leq \mathbb{P}(t|X_{1,i}| + (1 - t)|X_{0,i}| > R) \\
&\leq \mathbb{P}\left(|X_{0,i}| > \frac{R - 1}{1 - t}\right).
\end{aligned}
$$

Since $X_{0,i}$ is a standard Gaussian variable and thus sub-Gaussian with parameter 1 (Wainwright, 2019, Example 2.1), we have the following tail probability bound,

$$
\mathbb{P}\left(|X_{0,i}| > \frac{R - 1}{1 - t}\right) \leq 2\exp\left(-\frac{(R - 1)^2}{2(1 - t)^2}\right)
\tag{26}
$$

Combining (24), (25) and (26), we have

$$
\int_{\{\|x\|_\infty\}>R} \|\boldsymbol{v}^*(x,t)\|^2 \pi_t(d\boldsymbol{x}) \leq 4(d + 1)^{3/2}\exp\left(-\frac{(R - 1)^2}{4(1 - t)^2}\right).
$$

Let the right-hand side in the above inequality be smaller than $\varepsilon^2$, we have

$$R \geq 2(1-t)\left(2\log 2 + \frac{3}{2}\log(d+1) + 2\log\frac{1}{\varepsilon}\right)^{1/2} + 1.$$

So we can set $R = \mathcal{O}(\sqrt{\log\frac{d}{\varepsilon}})$ to guarantee $\|(\boldsymbol{v}^*(\boldsymbol{x},t) - \widehat{\boldsymbol{v}}_\theta(\boldsymbol{x},t))\mathbb{1}\{\|\boldsymbol{x}\|_\infty > R\}\|_{L^2(\pi_t)} \leq \varepsilon.$ $\qquad\square$

## B. Generalization Error

### B.1. Bounding loss function

**Lemma B.1.** *For any neural network $\boldsymbol{v}$ in $\mathrm{NN}(L, M, J, K, \kappa, \gamma_1, \gamma_2)$, we have $\sup_{\boldsymbol{x}\in[0,1]^d} |\ell(\boldsymbol{x},\boldsymbol{v})| \lesssim d + K^2$.*

*Proof.* Using the inequality $(a+b)^2 \leq 2a^2 + 2b^2$, we have

$$\begin{aligned}
\ell(\boldsymbol{x},\boldsymbol{v}) &= \frac{1}{T}\int_0^T \int \|\boldsymbol{x} - \boldsymbol{x}_0 - \boldsymbol{v}(t\boldsymbol{x} + (1-t)\boldsymbol{x}_0, t)\|^2 \pi_0(\mathrm{d}\boldsymbol{x}_0)\mathrm{d}t \\
&\lesssim \frac{1}{T}\int_0^T \int \|\boldsymbol{x} - \boldsymbol{x}_0\|^2 \pi_0(\mathrm{d}\boldsymbol{x}_0)\mathrm{d}t + \frac{1}{T}\int_0^T \int \|\boldsymbol{v}(t\boldsymbol{x} + (1-t)\boldsymbol{x}_0, t)\|^2 \pi_0(\mathrm{d}\boldsymbol{x}_0)\mathrm{d}t \\
&\lesssim d + \sup_{\boldsymbol{x},t}\|\boldsymbol{v}(\boldsymbol{x},t)\|^2 \\
&\lesssim d + K^2,
\end{aligned}$$

where the second inequality follows from the fact that $\pi_1$ is supported on $[0,1]^d$ and $\mathbb{E}[\|\boldsymbol{x}_0\|^2] = d$. This concludes the proof. $\qquad\square$

### B.2. Covering number evaluation

**Lemma B.2** (Covering number of $\mathcal{H}$). *For a neural network $\boldsymbol{v} : \mathbb{R}^d \times \mathbb{R} \to \mathbb{R}^d$, we define $\ell : \mathbb{R}^d \to \mathbb{R}$ as*

$$\ell(\boldsymbol{x},\boldsymbol{v}) := \frac{1}{T}\int_0^T \int \|\boldsymbol{x} - \boldsymbol{x}_0 - \boldsymbol{v}(t\boldsymbol{x} + (1-t)\boldsymbol{x}_0, t)\|^2 \pi_0(\boldsymbol{x}_0)\mathrm{d}\boldsymbol{x}_0\mathrm{d}t.$$

*For the hypotheses network class $\mathcal{V} = \mathrm{NN}(L, M, J, K, \kappa, \gamma_1, \gamma_2)$, we define a function class $\mathcal{H} := \{\ell(\cdot,\boldsymbol{v}) : \boldsymbol{v} \in \mathrm{NN}\}$.*

$$\log\mathcal{N}(\delta, \mathrm{NN}, \|\cdot\|_{L^\infty([-D,D]^d\times[0,1])}) \lesssim JL\log\left(\frac{LM(D\vee 1)\kappa}{\delta}\right), \tag{27}$$

*and based on this, the covering number of $\mathcal{H}$ is evaluated by*

$$\log\mathcal{N}(\delta, \mathcal{H}, \|\cdot\|_{L^\infty([0,1]^d)}) \lesssim JL\log\left(\frac{(K+d^{1/2})LM\kappa\sqrt{\log((K^2+d)/\delta)}}{\delta}\right). \tag{28}$$

*Proof.* The first bound (27) is directly obtain from (Chen et al., 2022b, Lemma 5.3), with a slight modification of the input region. The evaluation of the covering number of $\mathcal{H}$ proceeds by showing that a $\delta$-covering of NN induces a $C(\delta)$-covering of $\mathcal{H}$, where $C(\delta)$ is a function of $\delta$.

Assume that there are two neural networks $\boldsymbol{v}_1$ and $\boldsymbol{v}_2$ satisfying $\|\boldsymbol{v}_1 - \boldsymbol{v}_2\|_{L^\infty([-D,D]^d\times[0,1])} \leq \delta$, we want to proof that there is a function $C(\cdot)$, such that $\|\ell(\cdot,\boldsymbol{v}_1) - \ell(\cdot,\boldsymbol{v}_2)\|_{L^\infty([0,1]^d)} \leq C(\delta)$. $D$ will be determined later based on $\delta$. We rewrite $\ell(\boldsymbol{x},\boldsymbol{v})$ as follows:

$$\ell(\boldsymbol{x},\boldsymbol{v}) = \frac{1}{T}\int_0^T \int \|\boldsymbol{x} - \boldsymbol{x}_0\|^2 - 2(\boldsymbol{x} - \boldsymbol{x}_0)^T\boldsymbol{v}(t\boldsymbol{x} + (1-t)\boldsymbol{x}_0, t) + \|\boldsymbol{v}(t\boldsymbol{x} + (1-t)\boldsymbol{x}_0, t)\|^2 \pi_0(\mathrm{d}\boldsymbol{x}_0)\mathrm{d}t.$$

Then we have the following upper bound:

$$|\ell(\boldsymbol{x}, \boldsymbol{v}_1) - \ell(\boldsymbol{x}, \boldsymbol{v}_2)| \leq \underbrace{\frac{2}{T} \int_0^T \int \|\boldsymbol{x} - \boldsymbol{x}_0\| \cdot \|\boldsymbol{v}_1 - \boldsymbol{v}_2\| \pi_0(\mathrm{d}\boldsymbol{x}_0)\mathrm{d}t}_{(A)}$$

$$+ \underbrace{\frac{1}{T} \int_0^T \int \|\boldsymbol{v}_1 - \boldsymbol{v}_2\| \cdot \|\boldsymbol{v}_1 + \boldsymbol{v}_2\| \pi_0(\mathrm{d}\boldsymbol{x}_0)\mathrm{d}t}_{(B)},$$

where the inequality follows from Cauchy-Schwartz inequality and the identity $\|\boldsymbol{x}_1 - \boldsymbol{x}_2\|^2 = (\boldsymbol{x}_1 - \boldsymbol{x}_2)^T(\boldsymbol{x}_1 + \boldsymbol{x}_2)$. We omit the input of $\boldsymbol{v}_1$ and $\boldsymbol{v}_2$ for brevity, when there is no ambiguity.

**An upper bound for term (A).** The Cauchy-Schwartz inequality implies

$$\frac{1}{T} \int_0^T \int \|\boldsymbol{x} - \boldsymbol{x}_0\| \cdot \|\boldsymbol{v}_1 - \boldsymbol{v}_2\| \pi_0(\mathrm{d}\boldsymbol{x}_0)\mathrm{d}t \leq \left(\frac{1}{T} \int_0^T \int \|\boldsymbol{x} - \boldsymbol{x}_0\|^2 \pi_0(\mathrm{d}\boldsymbol{x}_0)\mathrm{d}t\right)^{1/2}$$

$$\cdot \left(\frac{1}{T} \int_0^T \int \|\boldsymbol{v}_1 - \boldsymbol{v}_2\|^2 \pi_0(\mathrm{d}\boldsymbol{x}_0)\mathrm{d}t\right)^{1/2}. \tag{29}$$

Note that $x \in [0,1]^d$ and $\boldsymbol{x}_0$ is a stand Gaussian variable, we have $\left(\frac{1}{T} \int_0^T \int \|\boldsymbol{x} - \boldsymbol{x}_0\|^2 \pi_0(\mathrm{d}\boldsymbol{x}_0)\mathrm{d}t\right)^{1/2} \lesssim d^{1/2}$. Using the change of variable $\boldsymbol{x}_t = t\boldsymbol{x} + (1-t)\boldsymbol{x}_0$, we have

$$\frac{1}{T} \int_0^T \int \|\boldsymbol{v}_1(t\boldsymbol{x} + (1-t)\boldsymbol{x}_0, t) - \boldsymbol{v}_2(t\boldsymbol{x} + (1-t)\boldsymbol{x}_0, t)\|^2 \pi_0(\mathrm{d}\boldsymbol{x}_0)\mathrm{d}t$$

$$= \frac{1}{T} \int_0^T \int \|\boldsymbol{v}_1(\boldsymbol{x}_t, t) - \boldsymbol{v}_2(\boldsymbol{x}_t, t)\|^2 \pi_{t|1}(\mathrm{d}\boldsymbol{x}_t|X_1 = \boldsymbol{x})\mathrm{d}t,$$

where $\pi_{t|1}$ is the distribution of $\boldsymbol{x}_t$ conditioned on $X_1 = \boldsymbol{x}$. We partition $\mathbb{R}^d$ into two subsets, $\{\boldsymbol{x}_t \in \mathbb{R}^d : \|\boldsymbol{x}_t\|_\infty \leq D\}$ and its complement $\{\boldsymbol{x}_t \in \mathbb{R}^d : \|\boldsymbol{x}_t\|_\infty > D\}$,

$$\int \|\boldsymbol{v}_1(\boldsymbol{x}_t, t) - \boldsymbol{v}_2(\boldsymbol{x}_t, t)\|^2 \pi_{t|1}(\mathrm{d}\boldsymbol{x}_t|X_1 = \boldsymbol{x}) = \int_{\{\|\boldsymbol{x}_t\|_\infty \leq D\}} \|\boldsymbol{v}_1(\boldsymbol{x}_t, t) - \boldsymbol{v}_2(\boldsymbol{x}_t, t)\|^2 \pi_{t|1}(\mathrm{d}\boldsymbol{x}_t|X_1 = \boldsymbol{x})$$

$$+ \int_{\{\|\boldsymbol{x}_t\|_\infty > D\}} \|\boldsymbol{v}_1(\boldsymbol{x}_t, t) - \boldsymbol{v}_2(\boldsymbol{x}_t, t)\|^2 \pi_{t|1}(\mathrm{d}\boldsymbol{x}_t|X_1 = \boldsymbol{x})$$

$$\lesssim \delta^2 + K^2 \mathbb{P}(\|t\boldsymbol{x} + (1-t)X_0\|_\infty > D).$$

Using the tail bound for Gaussian variable in (26), we obtain

$$\int \|\boldsymbol{v}_1(\boldsymbol{x}_t, t) - \boldsymbol{v}_2(\boldsymbol{x}_t, t)\|^2 \pi_{t|1}(\mathrm{d}\boldsymbol{x}_t|X_1 = \boldsymbol{x}) \lesssim \delta^2 + K^2 d \exp\left(-\frac{(D-1)^2}{2(1-t)^2}\right) \tag{30}$$

Combining (29) and (30), we get

$$\frac{1}{T} \int_0^T \int \|\boldsymbol{x} - \boldsymbol{x}_0\| \cdot \|\boldsymbol{v}_1 - \boldsymbol{v}_2\| \pi_0(\mathrm{d}\boldsymbol{x}_0)\mathrm{d}t \lesssim d^{1/2}\left(\delta^2 + K^2 d \exp\left(-\frac{(D-1)^2}{2(1-t)^2}\right)\right)^{1/2}$$

$$\lesssim d^{1/2}\left(\delta + K d^{1/2} \exp\left(-\frac{(D-1)^2}{4(1-t)^2}\right)\right) \tag{31}$$

$$\lesssim d^{1/2}\delta + Kd \exp\left(-\frac{(D-1)^2}{4}\right),$$

where the second inequality follows from the inequality $\sqrt{a+b} \leq \sqrt{a} + \sqrt{b}$, for $a \geq 0, b \geq 0$. The third inequality follows from the fact that $t \in [0, T]$.

**An upper bound for term (B).** Again, using Cauchy-Schwartz inequality, we have

$$
\frac{1}{T}\int_0^T \int \|\boldsymbol{v}_1 - \boldsymbol{v}_2\| \cdot \|\boldsymbol{v}_1 + \boldsymbol{v}_2\| \pi_0(\mathrm{d}\boldsymbol{x}_0)\mathrm{d}t \leq \left(\frac{1}{T}\int_0^T \int \|\boldsymbol{v}_1 - \boldsymbol{v}_2\|^2 \pi_0(\mathrm{d}\boldsymbol{x}_0)\mathrm{d}t\right)^{1/2}
$$

$$
\cdot \left(\frac{1}{T}\int_0^T \int \|\boldsymbol{v}_1 + \boldsymbol{v}_2\|^2 \pi_0(\mathrm{d}\boldsymbol{x}_0)\mathrm{d}t\right)^{1/2} \tag{32}
$$

$$
\lesssim K\left(\delta + Kd^{1/2}\exp\left(-\frac{(D-1)^2}{4}\right)\right)
$$

where the second inequality follows from the same argument in (30).

Combining (31) and (32), we obtain

$$
\sup_{\boldsymbol{x}\in[0,1]^d} |\ell(\boldsymbol{x}, \boldsymbol{v}_1) - \ell(\boldsymbol{x}, \boldsymbol{v}_2)| \lesssim (K + d^{1/2})\left(\delta + Kd^{1/2}\exp\left(-\frac{(D-1)^2}{4}\right)\right). \tag{33}
$$

Thus, a $\delta$-covering of NN w.r.t. $\|\cdot\|_{L^\infty([-D,D]^d \times [0,1])}$ induces a $C(K + \delta^{1/2})\left(\delta + Kd^{1/2}\exp(-(D-1)^2/4)\right)$-covering of $\mathcal{H}$, where $C$ is a universal constant. Let $Kd^{1/2}\exp(-(D-1)^2/4)$ be smaller than $\frac{\delta}{2C(K+d^{1/2})}$, we obtain $D \geq 2\sqrt{\log\frac{2CKd^{1/2}(K+d^{1/2})}{\delta}} + 1 =: D(\delta)$. Based on the above statements, a $\frac{\delta}{2C(K+d^{1/2})}$-covering of NN w.r.t. $\|\cdot\|_{L^\infty([-D(\delta),D(\delta)]^d \times [0,T])}$ induces a $\delta$-covering of $\mathcal{H}$.

Therefore, we obtain

$$
\log\mathcal{N}(\delta, \mathcal{H}, \|\cdot\|_{L^\infty([0,1]^d)}) \leq \log\mathcal{N}\left(\frac{\delta}{2C(K+d^{1/2})}, \mathrm{NN}, \|\cdot\|_{L^\infty([-D(\delta),D(\delta)]^d \times [0,T])}\right)
$$

$$
\lesssim JL\log\left(\frac{2C(K+d^{1/2})LMD(\delta)\kappa}{\delta}\right) \tag{34}
$$

$$
\lesssim JL\log\left(\frac{(K+d^{1/2})LM\kappa\sqrt{\log((K^2+d)/\delta)}}{\delta}\right).
$$

It concludes the proof. $\qquad\square$

### B.3. Proof of Theorem 4.4

*Proof of Theorem 4.4.* The generalization error $\mathcal{L}(\widehat{\boldsymbol{v}}) - \inf_{\boldsymbol{v}\in\mathrm{NN}}\mathcal{L}(\boldsymbol{v})$ can be decomposed into

$$
\mathcal{L}(\widehat{\boldsymbol{v}}) - \inf_{\boldsymbol{v}\in\mathrm{NN}}\mathcal{L}(\boldsymbol{v}) = \mathcal{L}(\widehat{\boldsymbol{v}}) - \widehat{\mathcal{L}}(\widehat{\boldsymbol{v}}) + \widehat{\mathcal{L}}(\widehat{\boldsymbol{v}}) - \widehat{\mathcal{L}}(\widetilde{\boldsymbol{v}}) + \widehat{\mathcal{L}}(\widetilde{\boldsymbol{v}}) - \mathcal{L}(\widetilde{\boldsymbol{v}})
$$

$$
\leq \mathcal{L}(\widehat{\boldsymbol{v}}) - \widehat{\mathcal{L}}(\widehat{\boldsymbol{v}}) + \widehat{\mathcal{L}}(\widetilde{\boldsymbol{v}}) - \mathcal{L}(\widetilde{\boldsymbol{v}}),
$$

where $\widetilde{\boldsymbol{v}} \in \mathrm{argmin}_{\boldsymbol{v}\in\mathrm{NN}}\mathcal{L}(\boldsymbol{v})$ and the last inequality follows from ERM.

For any $\boldsymbol{v}$, we have $\mathcal{L}(\boldsymbol{v}) - \widehat{\mathcal{L}}(\boldsymbol{v}) = \mathcal{L}(\boldsymbol{v}) - \overline{\mathcal{L}}(\boldsymbol{v}) + \overline{\mathcal{L}}(\boldsymbol{v}) - \widehat{\mathcal{L}}(\boldsymbol{v}) = \frac{1}{n}\sum_{i=1}^n (\mathcal{L}(\boldsymbol{v}) - \ell(\boldsymbol{x}_{1,i}, \boldsymbol{v})) + \frac{1}{n}\sum_{i=1}^n (\ell(\boldsymbol{x}_{1,i}, \boldsymbol{v}) - \widehat{\ell}(\boldsymbol{x}_{1,i}, \boldsymbol{v}))$, where the first term only involves sample from target distribution $\pi_1$ and the second term involves sample from $\mathrm{Unif}[0,T]$ and prior distribution $\pi_0$. Both of the two terms can be bounded by using a covering number argument.

●**Bounding** $\frac{1}{n}\sum_{i=1}^n (\mathcal{L}(\boldsymbol{v}) - \ell(\boldsymbol{x}_{1,i}, \boldsymbol{v}))$. Let $\{\ell_k\}_{k=1}^{N_1}$ be a $\tau$-covering of $\mathcal{H}$, where $N_1 = \mathcal{N}(\delta, \mathcal{H}, \|\cdot\|_{L^\infty([0,1]^d)})$. For every $\ell \in \mathcal{H}$, there exists a $k$, such that $\|\ell - \ell_k\|_{L^\infty([0,1]^d)} \leq \tau$. Thus, we have

$$
\frac{1}{n}\sum_{i=1}^n (\mathbb{E}[\ell(\boldsymbol{x})] - \ell(\boldsymbol{x}_{1,i})) \leq \frac{1}{n}\sum_{i=1}^n (\mathbb{E}[\ell_k(\boldsymbol{x})] - \ell_k(\boldsymbol{x}_{1,i})) + 2\tau
$$

$$
\leq \max_{k=1,\ldots,N_1} \frac{1}{n}\sum_{i=1}^n (\mathbb{E}[\ell_k(\boldsymbol{x})] - \ell_k(\boldsymbol{x}_{1,i})) + 2\tau. \tag{35}
$$

Take supremum over $\mathcal{H}$ on both sides, we get

$$\sup_{\ell \in \mathcal{H}} \frac{1}{n} \sum_{i=1}^{n} \left( \mathbb{E}[\ell(\boldsymbol{x})] - \ell(\boldsymbol{x}_{1,i}) \right) \leq \max_{k=1,\ldots,N_1} \frac{1}{n} \sum_{i=1}^{n} \left( \mathbb{E}[\ell_k(\boldsymbol{x})] - \ell_k(\boldsymbol{x}_{1,i}) \right) + 2\tau.$$

Thus, we have

$$\mathbb{P}\left( \sup_{\ell \in \mathcal{H}} \frac{1}{n} \left( \mathbb{E}[\ell(\boldsymbol{x})] - \ell(\boldsymbol{x}_{1,i}) \right) > \varepsilon + 2\tau \right) \leq \mathbb{P}\left( \max_{k=1,\ldots,N_1} \frac{1}{n} \sum_{i=1}^{n} \left( \mathbb{E}[\ell_k(\boldsymbol{x})] - \ell_k(\boldsymbol{x}_{1,i}) \right) > \varepsilon \right)$$

$$\leq \sum_{k=1}^{N_1} \mathbb{P}\left( \frac{1}{n} \sum_{i=1}^{n} \left( \mathbb{E}[\ell_k(\boldsymbol{x})] - \ell_k(\boldsymbol{x}_{1,i}) \right) > \varepsilon \right) \tag{36}$$

Invoking Lemma B.3, we get

$$\mathbb{P}\left( \sup_{\ell \in \mathcal{H}} \frac{1}{n} \left( \mathbb{E}[\ell(\boldsymbol{x})] - \ell(\boldsymbol{x}_{1,i}) \right) > \varepsilon + 2\tau \right) \leq N_1 \exp\left( -\frac{n\varepsilon^2}{2B^2} \right),$$

where $B = \mathcal{O}(d + K^2)$ by Lemma B.1. Letting $\varepsilon = \sqrt{\frac{2B^2 \log(N/\delta)}{n}}$, with probability of at least $1 - \delta$,

$$\frac{1}{n} \sum_{i=1}^{n} (\mathcal{L}(\boldsymbol{v}) - \ell(\boldsymbol{x}_{1,i}, \boldsymbol{v})) \leq \sup_{\ell \in \mathcal{H}} \frac{1}{n} \left( \mathbb{E}[\ell(\boldsymbol{x})] - \ell(\boldsymbol{x}_{1,i}) \right) \leq \sqrt{\frac{2B^2 \log(N_1/\delta)}{n}} + 2\tau.$$

•**Bounding** $\ell(\boldsymbol{x}_{1,i}, \boldsymbol{v}) - \widehat{\ell}(\boldsymbol{x}_{1,i}, \boldsymbol{v})$**.** We define $r((\boldsymbol{x}_0, t), \boldsymbol{v}, x) := \|\boldsymbol{x} - \boldsymbol{x}_0 - \boldsymbol{v}(t\boldsymbol{x} + (1 - t)\boldsymbol{x}_0, t)\|^2$ and its truncation $r^{\text{trunc}}((\boldsymbol{x}_0, t), \boldsymbol{v}, x) := \|\boldsymbol{x} - \boldsymbol{x}_0 - \boldsymbol{v}(t\boldsymbol{x} + (1 - t)\boldsymbol{x}_0, t)\|^2 \mathbb{1}\{\|\boldsymbol{x}_0\|_\infty \leq R\}$, where $R$ will be determined later depending on the tolerance. Thus, we have $\ell(\boldsymbol{x}_{1,i}, \boldsymbol{v}) = \mathbb{E}_{t, \boldsymbol{x}_0}[r((\boldsymbol{x}_0, t), \boldsymbol{v}, \boldsymbol{x}_{1,i})]$. We also define $\ell^{\text{trunc}}(\boldsymbol{x}, \boldsymbol{v}) := \mathbb{E}[r^{\text{trunc}}((\boldsymbol{x}_0, t), \boldsymbol{v}, x)]$ and its empirical version $\widehat{\ell}^{\text{trunc}}(\boldsymbol{x}, \boldsymbol{v}) = \frac{1}{m} \sum_{j=1}^{m} r^{\text{trunc}}((\boldsymbol{x}_{0,j}, t_j), \boldsymbol{v}, x)$ given $m$ i.i.d. sample $\{(\boldsymbol{x}_{0,j}, t_j)\}_{j=1}^{m}$ from $\pi_0$ and $\text{Unif}[0, T]$.

We have the following decomposition:

$$\ell(\boldsymbol{x}_{1,i}, \boldsymbol{v}) - \widehat{\ell}(\boldsymbol{x}_{1,i}, \boldsymbol{v}) = \underbrace{\ell(\boldsymbol{x}_{1,i}, \boldsymbol{v}) - \ell^{\text{trunc}}(\boldsymbol{x}_{1,i}, \boldsymbol{v})}_{\text{Truncation error (I)}} + \underbrace{\ell^{\text{trunc}}(\boldsymbol{x}_{1,i}, \boldsymbol{v}) - \widehat{\ell}^{\text{trunc}}(\boldsymbol{x}_{1,i}, \boldsymbol{v})}_{\text{Statistical error}}$$

$$+ \underbrace{\widehat{\ell}^{\text{trunc}}(\boldsymbol{x}_{1,i}, \boldsymbol{v}) - \widehat{\ell}(\boldsymbol{x}_{1,i}, \boldsymbol{v})}_{\text{Truncation error (II)}} \tag{37}$$

For Truncation error (I), we can use the concentration of Gaussian variables to control this term. For Statistical error, we can use a covering number argument to control it. Due to the concentration of Gaussian variables, Truncation error (II) is equal to zero with high probability.

We first control the first term. Using the Cauchy-Schwartz inequality and $(a + b)^2 \leq 2a^2 + 2b^2$, we have the following:

$$\text{Truncation error (I)} = \mathbb{E}_{t, \boldsymbol{x}_0}[\|\boldsymbol{x} - \boldsymbol{x}_0 - \boldsymbol{v}(t\boldsymbol{x} + (1 - t)\boldsymbol{x}_0, t)\|^2 \mathbb{1}\{\|\boldsymbol{x}_0\|_\infty > R\}]$$

$$\leq \mathbb{E}[\|\boldsymbol{x} - \boldsymbol{x}_0 - \boldsymbol{v}(t\boldsymbol{x} + (1 - t)\boldsymbol{x}_0, t)\|^4]^{1/2} \cdot \mathbb{P}(\|\boldsymbol{x}_0\|_\infty > R)^{1/2}$$

$$\leq 2 \left( \mathbb{E}[\|\boldsymbol{x} - \boldsymbol{x}_0\|^4] + \mathbb{E}[\|\boldsymbol{v}(t\boldsymbol{x} + (1 - t)\boldsymbol{x}_0, t)\|^4] \right)^{1/2} \cdot \mathbb{P}(\|\boldsymbol{x}_0\|_\infty > R)^{1/2}.$$

Note that $\mathbb{E}[\|\boldsymbol{x} - \boldsymbol{x}_0\|^4] \leq 4\|\boldsymbol{x}\|^4 + 4\mathbb{E}[\|\boldsymbol{x}_0\|^4] \leq 8d(d + 1)$ and $\sup_{\boldsymbol{x}, t} \|\boldsymbol{v}(\boldsymbol{x}, t)\| \leq K$ for any $\boldsymbol{v} \in \text{NN}$, we have

$$\text{Truncation error (I)} \leq 2 \left( 8d(d + 1) + K^4 \right)^{1/2} \cdot \mathbb{P}(\|\boldsymbol{x}_0\|_\infty > R)^{1/2}. \tag{38}$$

Denote the $k$-coordinate of $\boldsymbol{x}_0$ by $x_0^{(k)}$, we have the following upper bound for the tail probability:

$$
\begin{aligned}
\mathbb{P}(\|\boldsymbol{x}_0\|_\infty > R) &= \mathbb{P}\left(\max_{k=1,\ldots,d}|x_0^{(k)}| > R\right) \\
&= \mathbb{P}\left(\bigcup_{k=1}^d \{|x_0^{(k)}| > R\}\right) \\
&\overset{(i)}{\leq} \sum_{k=1}^d \mathbb{P}\left(|x_0^{(k)}| > R\right) \\
&\leq 2d\exp\left(-\frac{R^2}{2}\right),
\end{aligned}
\tag{39}
$$

where inequality (i) follows from union inequality. Combining (38) and (39), we obtain

$$
\text{Truncation error (I)} \leq 8(d^2(d+1) + dK^4)^{1/2} \cdot \exp\left(-\frac{R^2}{4}\right).
\tag{40}
$$

Next, we show that Truncation error (II) vanishes with high probability. Note that when $\|\boldsymbol{x}_{0,j}\| \leq R$ for $j = 1,\ldots,m$, Truncation error vanishes. It implies

$$
\begin{aligned}
\mathbb{P}(\widehat{\ell}^{\text{trunc}}(\boldsymbol{x}_{1,i}, \boldsymbol{v}) - \widehat{\ell}(\boldsymbol{x}_{1,i}, \boldsymbol{v}) = 0) &\geq \mathbb{P}\left(\bigcap_{j=1}^m \{\|\boldsymbol{x}_{0,j}\| \leq R\}\right) \\
&= 1 - \mathbb{P}\left(\bigcup_{j=1}^m \{\|\boldsymbol{x}_{0,j}\| > R\}\right) \\
&\geq 1 - md\exp\left(-\frac{R^2}{2}\right)
\end{aligned}
\tag{41}
$$

Finally, we control the Statistical error. Note that, for $R > 1$, we have $t\boldsymbol{x} + (1-t)\boldsymbol{x}_0 \in [-R, R]^d$ for all $\|\boldsymbol{x}_0\|_\infty \leq R$, since $\|\boldsymbol{x}\|_\infty \leq 1$ with probability 1. Given a $\frac{\tau}{2\sqrt{d}(R+1)+2K}$-covering $\{\boldsymbol{v}_i\}_{i=1}^{N_2}$ of NN, w.r.t. $\|\cdot\|_{L^\infty([-R,R]^d \times [0,T])}$, where $N_2 = \mathcal{N}(\frac{\tau}{2\sqrt{d}(R+1)+2K}, \text{NN}, \|\cdot\|_{L^\infty([-R,R]^d \times [0,T])})$. For any $\boldsymbol{v} \in \text{NN}$, there exists $k = 1,\ldots,N_2$, such that $\|\boldsymbol{v} - \boldsymbol{v}_k\|_{L^\infty([-R,R]^d \times [0,T])} \leq \frac{\tau}{2\sqrt{d}(R+1)+2K}$. For any $\boldsymbol{x}_0 \in [-R, R]^d$ and $t \in [0, T]$, we have the following bound for $|r^{\text{trunc}}((\boldsymbol{x}_0, t), \boldsymbol{v}_k, x) - r^{\text{trunc}}((\boldsymbol{x}_0, t), \boldsymbol{v}, x)|$:

$$
\begin{aligned}
|r^{\text{trunc}}((\boldsymbol{x}_0, t), \boldsymbol{v}_k, x) - r^{\text{trunc}}((\boldsymbol{x}_0, t), \boldsymbol{v}, x)| &\leq |\langle \boldsymbol{v}_k - \boldsymbol{v}, 2\boldsymbol{x} - 2\boldsymbol{x}_0 - \boldsymbol{v}_k - \boldsymbol{v}\rangle| \\
&\leq \|\boldsymbol{v} - \boldsymbol{v}_k\|_{L^\infty([-R,R]^d \times [0,T])} \\
&\quad \cdot (2\|\boldsymbol{x}\| + 2\|\boldsymbol{x}_0\| + \|\boldsymbol{v}_1\| + \|\boldsymbol{v}_2\|) \\
&\leq (2\sqrt{d}(R+1) + 2K)\frac{\tau}{2\sqrt{d}(R+1)+2K} = \tau.
\end{aligned}
$$

For any $\boldsymbol{v} \in \text{NN}$, we have

$$
\begin{aligned}
&\ell^{\text{trunc}}(\boldsymbol{x}_{1,i}, \boldsymbol{v}) - \widehat{\ell}^{\text{trunc}}(\boldsymbol{x}_{1,i}, \boldsymbol{v}) \\
=& \mathbb{E}_{\boldsymbol{x}_0, t}[r^{\text{trunc}}((\boldsymbol{x}_0, t), \boldsymbol{v}, \boldsymbol{x}_{1,i})] - \frac{1}{m}\sum_{j=1}^m r^{\text{trunc}}((\boldsymbol{x}_{0,j}, t_j), \boldsymbol{v}, \boldsymbol{x}_{1,i}) \\
\leq& \mathbb{E}_{\boldsymbol{x}_0, t}[r^{\text{trunc}}((\boldsymbol{x}_0, t), \boldsymbol{v}_k, \boldsymbol{x}_{1,i})] - \frac{1}{m}\sum_{j=1}^m r^{\text{trunc}}((\boldsymbol{x}_{0,j}, t_j), \boldsymbol{v}_k, \boldsymbol{x}_{1,i}) + 2\tau \\
\leq& \max_{k=1,\ldots,N_2}\left\{\mathbb{E}_{\boldsymbol{x}_0, t}[r^{\text{trunc}}((\boldsymbol{x}_0, t), \boldsymbol{v}_k, \boldsymbol{x}_{1,i})] - \frac{1}{m}\sum_{j=1}^m r^{\text{trunc}}((\boldsymbol{x}_{0,j}, t_j), \boldsymbol{v}_k, \boldsymbol{x}_{1,i})\right\} + 2\tau.
\end{aligned}
$$

Taking supremum on both sides, we obtain

$$\sup_{\boldsymbol{v} \in \mathrm{NN}} \left\{ \ell^{\mathrm{trunc}}(\boldsymbol{x}_{1,i}, \boldsymbol{v}) - \widehat{\ell}^{\mathrm{trunc}}(\boldsymbol{x}_{1,i}, \boldsymbol{v}) \right\}$$

$$\leq 2\tau + \max_{k=1,\ldots,N_2} \left\{ \mathbb{E}_{\boldsymbol{x}_0,t}[r^{\mathrm{trunc}}((\boldsymbol{x}_0,t), \boldsymbol{v}_k, \boldsymbol{x}_{1,i})] - \frac{1}{m}\sum_{j=1}^{m} r^{\mathrm{trunc}}((\boldsymbol{x}_{0,j}, t_j), \boldsymbol{v}_k, \boldsymbol{x}_{1,i}) \right\}.$$

Thus, we have

$$\mathbb{P}\left( \sup_{\boldsymbol{v} \in \mathrm{NN}} \left\{ \ell^{\mathrm{trunc}}(\boldsymbol{x}_{1,i}, \boldsymbol{v}) - \widehat{\ell}^{\mathrm{trunc}}(\boldsymbol{x}_{1,i}, \boldsymbol{v}) \right\} > \varepsilon + 2\tau \right)$$

$$\leq \mathbb{P}\left( \max_{k=1,\ldots,N_2} \left\{ \mathbb{E}_{\boldsymbol{x}_0,t}[r^{\mathrm{trunc}}((\boldsymbol{x}_0,t), \boldsymbol{v}_k, \boldsymbol{x}_{1,i})] - \frac{1}{m}\sum_{j=1}^{m} r^{\mathrm{trunc}}((\boldsymbol{x}_{0,j}, t_j), \boldsymbol{v}_k, \boldsymbol{x}_{1,i}) \right\} > \varepsilon \right)$$

$$= \mathbb{P}\left( \bigcup_{k=1}^{N_2} \left\{ \mathbb{E}_{\boldsymbol{x}_0,t}[r^{\mathrm{trunc}}((\boldsymbol{x}_0,t), \boldsymbol{v}_k, \boldsymbol{x}_{1,i})] - \frac{1}{m}\sum_{j=1}^{m} r^{\mathrm{trunc}}((\boldsymbol{x}_{0,j}, t_j), \boldsymbol{v}_k, \boldsymbol{x}_{1,i}) > \varepsilon \right\} \right) \tag{42}$$

$$\overset{(i)}{\leq} \sum_{k=1}^{N_2} \mathbb{P}\left( \mathbb{E}_{\boldsymbol{x}_0,t}[r^{\mathrm{trunc}}((\boldsymbol{x}_0,t), \boldsymbol{v}_k, \boldsymbol{x}_{1,i})] - \frac{1}{m}\sum_{j=1}^{m} r^{\mathrm{trunc}}((\boldsymbol{x}_{0,j}, t_j), \boldsymbol{v}_k, \boldsymbol{x}_{1,i}) > \varepsilon \right),$$

where inequality (i) follows from the union inequality. Note that $0 \leq r^{\mathrm{trunc}}((\boldsymbol{x}_0,t), \boldsymbol{v}_k, \boldsymbol{x}_{1,i}) \leq 2d(R+1)^2 + 2K^2$, applying Lemma B.3, we obtain

$$\mathbb{P}\left( \mathbb{E}_{\boldsymbol{x}_0,t}[r^{\mathrm{trunc}}((\boldsymbol{x}_0,t), \boldsymbol{v}_k, \boldsymbol{x}_{1,i})] - \frac{1}{m}\sum_{j=1}^{m} r^{\mathrm{trunc}}((\boldsymbol{x}_{0,j}, t_j), \boldsymbol{v}_k, \boldsymbol{x}_{1,i}) > \varepsilon \right) \leq \exp\left( -\frac{m\varepsilon^2}{8(d(R+1)^2 + K^2)^2} \right). \tag{43}$$

Combining Equation (42) and Equation (43), we obtain

$$\mathbb{P}\left( \sup_{\boldsymbol{v} \in \mathrm{NN}} \left\{ \ell^{\mathrm{trunc}}(\boldsymbol{x}_{1,i}, \boldsymbol{v}) - \widehat{\ell}^{\mathrm{trunc}}(\boldsymbol{x}_{1,i}, \boldsymbol{v}) \right\} > \varepsilon + 2\tau \right) \leq N_2 \exp\left( -\frac{m\varepsilon^2}{8(d(R+1)^2 + K^2)^2} \right). \tag{44}$$

Let $N_2 \exp\left( -\frac{n\varepsilon^2}{8(d(R+1)^2 + K^2)^2} \right) = \delta/2$, we have, with probability of at least $1 - \delta/2$,

$$\sup_{\boldsymbol{v} \in \mathrm{NN}} \left\{ \ell^{\mathrm{trunc}}(\boldsymbol{x}_{1,i}, \boldsymbol{v}) - \widehat{\ell}^{\mathrm{trunc}}(\boldsymbol{x}_{1,i}, \boldsymbol{v}) \right\} \leq 2\tau + \sqrt{\frac{8(d(R+1)^2 + K^2)^2 \log(2N_2/\delta)}{m}} \tag{45}$$

Let $md \exp\left( -\frac{R^2}{2} \right) = \frac{\delta}{2}$ in (41), we have, with probability of at least $1 - \delta/2$,

$$\widehat{\ell}^{\mathrm{trunc}}(\boldsymbol{x}_{1,i}, \boldsymbol{v}) - \widehat{\ell}(\boldsymbol{x}_{1,i}, \boldsymbol{v}) = 0. \tag{46}$$

Combining (40), (45), and (46), we have, with probability of at least $1 - \delta$,

$$\sup_{\boldsymbol{v} \in \mathrm{NN}} \ell(\boldsymbol{x}_{1,i}, \boldsymbol{v}) - \widehat{\ell}(\boldsymbol{x}_{1,i}, \boldsymbol{v}) \leq 8 \left( \frac{\delta(d(d+1) + K^4)}{2m} \right)^{1/2} + 2\tau$$

$$+ \sqrt{\frac{8(d(\sqrt{2\log(2md/\delta)} + 1)^2 + K^2)^2 \log(2N_2/\delta)}{m}}$$

Combining the bounds for $\frac{1}{n}\sum_{i=1}^{n}(\mathcal{L}(\boldsymbol{v}) - \ell(\boldsymbol{x}_{1,i}, \boldsymbol{v}))$ and $\ell(\boldsymbol{x}_{1,i}, \boldsymbol{v}) - \widehat{\ell}(\boldsymbol{x}_{1,i}, \boldsymbol{v})$, we otbain, with probability of at least $1 - 4\delta$,

$$\mathcal{L}(\widehat{\boldsymbol{v}}) - \inf_{\boldsymbol{v} \in \mathrm{NN}} \mathcal{L}(\boldsymbol{v}) = \mathcal{O}\left( \tau + \frac{\delta^{1/2}(d+1) + K^2}{\sqrt{m}} + \frac{(d\log(md/\delta) + K^2) \cdot \sqrt{\log(N_2/\delta)}}{\sqrt{m}} \right.$$

$$\left. + \frac{(d + K^2)\sqrt{\log(N_1/\delta)}}{\sqrt{n}} \right).$$

• **Balancing error terms.** By choosing NN as in Theorem 3.1 with approximation error $\varepsilon$, we have $\inf_{\boldsymbol{v} \in \text{NN}} \mathcal{L}(\boldsymbol{v}) - \mathcal{L}(\boldsymbol{v}^*) \leq (\sqrt{d} + 1)\varepsilon$. Setting $\delta = \frac{1}{4n}$ and $\tau = \frac{1}{n}$ gives rise to

$$\frac{1}{T} \int_0^T \|\widehat{\boldsymbol{v}}(\cdot, t) - \boldsymbol{v}^*(\cdot, t)\|_{L^2(\pi_t)}^2 \mathrm{d}t = \widetilde{\mathcal{O}}\left(\frac{1}{(1-T)^4}\left(\frac{1}{n} + \frac{1}{\sqrt{mn}} + \frac{(1/\varepsilon)^{\frac{d+1}{2}}}{\sqrt{m}} + \frac{(1/\varepsilon)^{\frac{d+1}{2}}}{\sqrt{n}} + \varepsilon^2\right)\right),$$

with probability of at least $1 - \frac{1}{n}$, where we omit factors in $d, \log n, \log m, \log(1 - T)$. By setting $\varepsilon = n^{-\frac{1}{d+5}}$, it holds

$$\frac{1}{T} \int_0^T \|\widehat{\boldsymbol{v}}(\cdot, t) - \boldsymbol{v}^*(\cdot, t)\|_{L^2(\pi_t)}^2 \mathrm{d}t = \widetilde{\mathcal{O}}\left(\frac{1}{(1-T)^4}\left(n^{-\frac{2}{d+5}} + n^{\frac{d+1}{2(d+5)}} m^{-\frac{1}{2}}\right)\right),$$

with probabiliy of at least $1 - \frac{1}{n}$. $\qquad\square$

## B.4. Auxiliary lemma

**Lemma B.3.** *Let $\mathcal{G}$ be a bounded function class, i.e., there exists a constant $B$ such that for any $g \in \mathcal{G}$ and any $\boldsymbol{x}$ in its domain, $0 \leq g(\boldsymbol{x}) \leq B$. Let $X_1, \ldots, X_n \in \mathbb{R}^d$ be i.i.d. random variables. For any $\delta \in (0, 1)$ and $g \in \mathcal{G}$, we have*

$$\mathbb{P}\left(\frac{1}{n}\sum_{i=1}^n (g(X_i) - \mathbb{E}[g(X)]) > \varepsilon\right) \leq \exp\left(-\frac{n\varepsilon^2}{2B^2}\right) \quad and \tag{47}$$

$$\mathbb{P}\left(\frac{1}{n}\sum_{i=1}^n (\mathbb{E}[g(X)] - g(X_i)) > \varepsilon\right) \leq \exp\left(-\frac{n\varepsilon^2}{2B^2}\right). \tag{48}$$

*Proof.* We first compute the moment generating function of $\frac{1}{n}\sum_{i=1}^n (\mathbb{E}[g(X)] - g(X_i))$,

$$\mathbb{E}\left[\exp\left(\frac{\lambda}{n}\sum_{i=1}^n (g(X_i) - E[g(X)])\right)\right] = \left(\mathbb{E}\left[\exp\left(\frac{\lambda}{n}(g(X_1) - E[g(X)])\right)\right]\right)^n, \tag{49}$$

the identity follows from the fact that $\boldsymbol{x}_1, \ldots, \boldsymbol{x}_n$ are i.i.d. random variables. Now, we try to upper bound $\mathbb{E}\left[\exp\left(\frac{\lambda}{n}(g(\boldsymbol{x}) - E[g(\boldsymbol{x})])\right)\right]$. Given an independent copy $X_1'$ of $X_1$, we have

$$\mathbb{E}\left[\exp\left(\frac{\lambda}{n}(g(X_1) - E[g(X)])\right)\right] = \mathbb{E}\left[\exp\left(\frac{\lambda}{n}(g(X_1) - E_{X_1'}[g(X_1')])\right)\right]$$
$$\leq \mathbb{E}_{X_1, X_1'}\left[\exp\left(\frac{\lambda}{n}(g(X_1) - g(X_1'))\right)\right] \tag{50}$$

Letting $\sigma$ be an independent Rademacher variable, note that the distribution of $(X_1 - X_1')$ is the same as that of $\sigma(X_1 - X_1')$, so that we have

$$\mathbb{E}_{X_1, X_1'}\left[\exp\left(\frac{\lambda}{n}(g(X_1) - g(X_1'))\right)\right] = \mathbb{E}_{X_1, X_1'}\left[\mathbb{E}_\sigma\left[\exp\left(\frac{\lambda}{n}\sigma(g(X_1) - g(X_1'))\right)\right]\right]$$
$$\overset{(i)}{\leq} \mathbb{E}_{X_1, X_1'}\left[\exp\left(\frac{\lambda^2(g(X_1) - g(X_1'))^2}{2n^2}\right)\right],$$

where (i) follows from Lemma B.4, applied conditionally with $(X_1, X_1')$ held fixed. Since $|g(X_1) - g(X_1')| \leq B$, we are guaranteed that

$$\mathbb{E}_{X_1, X_1'}\left[\exp\left(\frac{\lambda^2(g(X_1) - g(X_1'))^2}{2n^2}\right)\right] \leq \exp\left(\frac{\lambda^2 B^2}{2n^2}\right) \tag{51}$$

Combining (49) and (51), we obtain,

$$\mathbb{E}\left[\exp\left(\frac{\lambda}{n}\sum_{i=1}^n (g(X_i) - E[g(X)])\right)\right] \leq \exp\left(\frac{\lambda^2 B^2}{2n}\right).$$

Using the Markov inequality, we have

$$\mathbb{P}\left(\frac{1}{n}\sum_{i=1}^{n}(g(X_i) - \mathbb{E}[g(X)]) > \varepsilon\right) \leq \exp\left(\frac{\lambda^2 B^2}{2n} - \lambda\varepsilon\right).$$

Let $\lambda = \frac{n\varepsilon}{B^2}$, we get the first inequality. The second inequality can be proved in the exact same argument. □

**Lemma B.4.** *Given a Rademacher random variable $\sigma$ takes the values $\{-1, 1\}$ equiprobably. We have, for any $\lambda \in \mathbb{R}$, $\mathbb{E}[e^{\lambda\sigma}] \leq e^{\lambda^2/2}$.*

*Proof.* By taking expectations and using the power-series expansion for the exponential, we obtain

$$\begin{aligned}
\mathbb{E}[e^{\lambda\sigma}] = \frac{1}{2}[e^{-\lambda} + e^{\lambda}] &= \frac{1}{2}\left[\sum_{k=0}^{\infty}\frac{(-\lambda)^k}{k!} + \sum_{k=0}^{\infty}\frac{(\lambda)^k}{k!}\right] \\
&= \sum_{k=0}^{\infty}\frac{\lambda^{2k}}{(2k)!} \\
&\leq 1 + \sum_{k=1}^{\infty}\frac{\lambda^{2k}}{2^k k!} \\
&= e^{\lambda^2/2}.
\end{aligned} \tag{52}$$

It concludes the proof. □

## C. Discretization Analysis

### C.1. Estimation Error

Consider the target continuous flow:

$$dX_t(\boldsymbol{x}) = \boldsymbol{v}^*(X_t(\boldsymbol{x}), t)dt, \ X_0(\boldsymbol{x}) = \boldsymbol{x} \sim \pi_0, \ 0 \leq t \leq T, \tag{53}$$

and the estimated continuous flow

$$d\widehat{X}_t(\boldsymbol{x}) = \widehat{\boldsymbol{v}}(\widehat{X}_t(\boldsymbol{x}), t)dt, \ \widehat{X}_0(\boldsymbol{x}) = \boldsymbol{x} \sim \pi_0, \ 0 \leq t \leq T. \tag{54}$$

Denote the distribution of $X_t(\boldsymbol{x})$ and $\widehat{X}_t(\boldsymbol{x})$ by $\pi_t$ and $\widehat{\pi}_t$, respectively. We have the following estimate of the Wasserstein-2 distance $W_2(\pi_T, \widehat{\pi}_T)$.

**Proposition C.1.** *Suppose Assumption 1.1 holds. For any velocity field $\boldsymbol{v}^*$ with Lipschitz constant $\zeta$ w.r.t. $\boldsymbol{x}$, given $n$ samples $\{\boldsymbol{x}_{1,i}\}_{i=1}^{n}$ from $\pi_1$ and $m$ samples from $\pi_0$ and $\mathrm{Unif}[0, T]$, we choose NN as in Theorem 3.1 with $\varepsilon = n^{-\frac{1}{d+5}}$. Then with probability of at least $1 - \frac{1}{n}$, it holds*

$$W_2(\pi_T, \widehat{\pi}_T) = \widetilde{\mathcal{O}}\left(e^{\gamma_1}\frac{\zeta^{d/4}}{(1-T)^2}n^{-\frac{1}{d+5}}\right). \tag{55}$$

The proof can be found in Proposition 5.1.

### C.2. Discretization Error

Now we consider the gap between estimated continuous flow and its discretization:

$$\begin{aligned}
d\widehat{X}_t(\boldsymbol{x}) &= \widehat{\boldsymbol{v}}(\widehat{X}_t(\boldsymbol{x}), t)dt, \ \widehat{X}_0(\boldsymbol{x}) = \boldsymbol{x} \sim \pi_0, \ 0 \leq t \leq T, \\
d\widetilde{X}_t(\boldsymbol{x}) &= \widehat{\boldsymbol{v}}(\widetilde{X}_{t_k}(\boldsymbol{x}), t_k)dt, \ t_k \leq t \leq t_{k+1}, k = 0, 1, \ldots, N-1, \ \widetilde{X}_0(\boldsymbol{x}) = \boldsymbol{x} \sim \pi_0.
\end{aligned}$$

Denote the distribution of $\widehat{X}_t(\boldsymbol{x})$ and $\widetilde{X}_t(\boldsymbol{x})$ by $\widehat{\pi}_t$ and $\widetilde{\pi}_t$, respectively.

**Lemma C.2.** *Let* $0 = t_0 < t_1 < \cdots < t_N = T$ *be the discretization points. For any neural network* $\widehat{v}$ *in* $\mathrm{NN}(L, M, J, K, \kappa, \gamma_1, \gamma_2)$, *we have:*

$$W_2(\widehat{\pi}_T, \widetilde{\pi}_T) = \mathcal{O}\left(e^{\gamma_1}(\gamma_1 K + \gamma_2)\sqrt{\sum_{k=0}^{N-1}(t_{k+1} - t_k)^3}\right),$$

*where* $\widehat{\pi}$ *is the distribution of the final output of the estimated sampling dynamics* (17).

*Proof.* By the same argument as in the proof of Proposition C.1, we have

$$W_2^2(\widehat{\pi}_t, \widetilde{\pi}_t) \leq \int_{R^d} \|\widehat{X}_t(\boldsymbol{x}) - \widetilde{X}_t(\boldsymbol{x})\|^2 \pi_0(\boldsymbol{x})\mathrm{d}\boldsymbol{x}.$$

Now, we consider the evolution of

$$L_t := \int_{\mathbb{R}^d} \|\widehat{X}_t(\boldsymbol{x}) - \widetilde{X}_t(\boldsymbol{x})\|^2 \pi_0(\boldsymbol{x})\mathrm{d}\boldsymbol{x}.$$

Since $\widetilde{X}_t(\boldsymbol{x})$ is piece-wise linear, we consider the evolution of $L_t$ on each split interval $[t_k, t_{k+1}]$. On interval $[t_k, t_{k+1}]$, we have

$$\frac{\mathrm{d}L_t}{\mathrm{d}t} = \int_{\mathbb{R}^d} 2\langle \widehat{v}_t(\widehat{X}_t(\boldsymbol{x})) - \widehat{v}_{t_k}(\widetilde{X}_{t_k}(\boldsymbol{x})), \widehat{X}_t(\boldsymbol{x}) - \widetilde{X}_t(\boldsymbol{x})\rangle \pi_0(\boldsymbol{x})\mathrm{d}\boldsymbol{x} \tag{56}$$

$$= \int_{\mathbb{R}^d} 2\langle \widehat{v}_t(\widehat{X}_t(\boldsymbol{x})) - \widehat{v}_t(\widetilde{X}_t(\boldsymbol{x})), \widehat{X}_t(\boldsymbol{x}) - \widetilde{X}_t(\boldsymbol{x})\rangle \pi_0(\boldsymbol{x})\mathrm{d}\boldsymbol{x} \tag{57}$$

$$+ \int_{\mathbb{R}^d} 2\langle \widehat{v}_t(\widetilde{X}_t(\boldsymbol{x})) - \widehat{v}_t(\widetilde{X}_{t_k}(\boldsymbol{x})), \widehat{X}_t(\boldsymbol{x}) - \widetilde{X}_t(\boldsymbol{x})\rangle \pi_0(\boldsymbol{x})\mathrm{d}\boldsymbol{x} \tag{58}$$

$$+ \int_{\mathbb{R}^d} 2\langle \widehat{v}_t(\widetilde{X}_{t_k}(\boldsymbol{x})) - \widehat{v}_{t_k}(\widetilde{X}_{t_k}(\boldsymbol{x})), \widehat{X}_t(\boldsymbol{x}) - \widetilde{X}_t(\boldsymbol{x})\rangle \pi_0(\boldsymbol{x})\mathrm{d}\boldsymbol{x} \tag{59}$$

For (57), by Cauchy-Schwartz inequality and the fact that $\widehat{v}$ is $\gamma_1$-Lipschitz continuous w.r.t. $\boldsymbol{x}$, we get

$$\int_{\mathbb{R}^d} 2\langle \widehat{v}_t(\widehat{X}_t(\boldsymbol{x})) - \widehat{v}_t(\widetilde{X}_t(\boldsymbol{x})), \widehat{X}_t(\boldsymbol{x}) - \widetilde{X}_t(\boldsymbol{x})\rangle \pi_0(\boldsymbol{x})\mathrm{d}\boldsymbol{x} \leq 2\gamma_1 \int_{R^d} \|\widehat{X}_t(\boldsymbol{x}) - \widetilde{X}_t(\boldsymbol{x})\|^2 \pi_0(\boldsymbol{x})\mathrm{d}\boldsymbol{x}. \tag{60}$$

For (58), note that $\widetilde{X}_t(\boldsymbol{x}) = \widetilde{X}_{t_k}(\boldsymbol{x}) + (t - t_k)\widehat{v}_{t_k}(\widetilde{X}_{t_k}(\boldsymbol{x}))$, we use the inequality $2\langle a, b\rangle \leq \|a\|^2 + \|b\|^2$ and the fact that $\widehat{v}$ is $\gamma_1$-Lipschitz continuous w.r.t. $\boldsymbol{x}$ to get

$$\int_{\mathbb{R}^d} 2\langle \widehat{v}_t(\widetilde{X}_t(\boldsymbol{x})) - \widehat{v}_t(\widetilde{X}_{t_k}(\boldsymbol{x})), \widehat{X}_t(\boldsymbol{x}) - \widetilde{X}_t(\boldsymbol{x})\rangle \pi_0(\boldsymbol{x})\mathrm{d}\boldsymbol{x}$$
$$\leq \int_{\mathbb{R}^d} \|\widehat{v}_t(\widetilde{X}_t(\boldsymbol{x})) - \widehat{v}_t(\widetilde{X}_{t_k}(\boldsymbol{x}))\|^2 \pi_0(\boldsymbol{x})\mathrm{d}\boldsymbol{x} + \int_{\mathbb{R}^d} \|\widehat{X}_t(\boldsymbol{x}) - \widetilde{X}_t(\boldsymbol{x})\|^2 \pi_0(\boldsymbol{x})\mathrm{d}\boldsymbol{x} \tag{61}$$
$$\leq \gamma_1^2(t - t_k)^2 \|\widehat{v}\|_{L^\infty}^2 + L_t$$
$$\leq \gamma_1^2(t - t_k)^2 K^2 + L_t,$$

where $K$ is the parameter of the neural networks in Theorem 4.4. For (59), the fact that $\widehat{v}$ is $\gamma_2$-Lipschitz continuous w.r.t. $t$ implies

$$\int_{\mathbb{R}^d} 2\langle \widehat{v}_t(\widetilde{X}_{t_k}(\boldsymbol{x})) - \widehat{v}_{t_k}(\widetilde{X}_{t_k}(\boldsymbol{x})), \widehat{X}_t(\boldsymbol{x}) - \widetilde{X}_t(\boldsymbol{x})\rangle \pi_0(\boldsymbol{x})\mathrm{d}\boldsymbol{x}$$
$$\leq \int_{\mathbb{R}^d} \|\widehat{X}_t(\boldsymbol{x}) - \widetilde{X}_t(\boldsymbol{x})\|^2 \pi_0(\boldsymbol{x})\mathrm{d}\boldsymbol{x} + \gamma_2^2(t - t_k)^2. \tag{62}$$

Combining (60), (61) and (62), we obtain

$$\frac{dL_t}{dt} \leq (2\gamma_1 + 2)L_t + (\gamma_1^2 K^2 + \gamma_2^2)(t - t_k)^2, \quad \text{on } [t_k, t_{k+1}].$$

Again, by Lemma C.6, we obtain

$$e^{-(2\gamma_1+2)t_{k+1}}L_{t_{k+1}} - e^{-(2\gamma_1+2)t_k}L_{t_k} \le \frac{1}{3}(\gamma_1^2 K^2 + \gamma_2^2)(t_{k+1} - t_k)^3.$$

Summing over $k$ and noting that $t_N = T$, we get

$$L_T \le \frac{1}{3}e^{2(\gamma_1+1)T}(\gamma_1^2 K^2 + \gamma_2^2)\sum_{k=0}^{N-1}(t_{k+1} - t_k)^3.$$

Thus, we have

$$W_2(\widehat{\pi}_T, \widetilde{\pi}_T) = \mathcal{O}\left(e^{\gamma_1}(\gamma_1 K + \gamma_2)\sqrt{\sum_{k=0}^{N-1}(t_{k+1} - t_k)^3}\right).$$

$\square$

**Lemma C.3.** *Suppose Assumption 1.1 holds, we have*

$$W_2(\pi_T, \pi_1) \lesssim (1-T)\sqrt{d}.$$

*Proof.* We consider the error from early stopping. Note that $X_T$ and $X_1$ form a coupling of $\pi_T$ and $\pi_1$, by the definition of Wasserstein-2 distance, we obtain

$$W_2(\pi_T, \pi_1) \le \mathbb{E}[\|X_T - X_1\|^2]^{1/2} \le (1-T)\mathbb{E}[\|X_1 - X_0\|^2]^{1/2}.$$

Since we assume $\pi_1$ is supported on $[0,1]^d$ and $\mathbb{E}[\|X_0\|^2] = d$, we have $W_2(\pi_T, \pi_1) \lesssim (1-T)\sqrt{d}$. $\square$

### C.3. Proof of Main Results

**Theorem C.4.** *Suppose Assumption 1.1 holds. Given $n$ samples from target distribution $\pi_1$ and the networks as in Theorem 4.4, with parameter $\zeta$ replaced by $\frac{d}{(1-T)^3}$, we use the estimated velocity field in (11), to generate samples and choose the maximal step size $\max_{k=0,1...,N-1}|t_{k+1} - t_k| = \mathcal{O}(n^{-\frac{1}{d+5}})$ and early stopping time $T(n) = 1 - (\log n)^{-1/6}$, we have*

$$W_2(\widetilde{\pi}_{T(n)}, \pi_1) \to 0, \quad \text{in probability.}$$

*Proof.* Lemma 1.3 shows that the velocity field $\boldsymbol{v}^*$ is $\frac{d}{(1-T)^3}$-Lipschitz continuous w.r.t. $\boldsymbol{x}$ on $\mathbb{R}^d \times [0,T]$, when $\frac{1}{2} < T < 1$. Combining Proposition C.1, Lemma C.2 and Lemma C.3, we obtain

$$W_2(\widetilde{\pi}_T, \pi_1) = \widetilde{\mathcal{O}}\left((1-T) + e^{\gamma_1}(\gamma_1 K + \gamma_2)\sqrt{\sum_{k=0}^{N-1}(t_{k+1} - t_k)^3} + e^{\gamma_1}\frac{\zeta^{d/4}}{(1-T)^2}n^{-\frac{1}{d+5}}\right).$$

By the choice of neural networks, we have $\gamma_1 = \mathcal{O}\left(\frac{10d^2}{(1-T)^3}\right)$. Letting $\max_{k=0,1...,N-1}|t_{k+1} - t_k| = \mathcal{O}(n^{-\frac{1}{d+5}})$, $T(n) = 1 - (\log n)^{-1/6}$ and omitting polynomials of logarithm, we obtain,

$$W_2(\widetilde{\pi}_T, \pi_1) = \widetilde{\mathcal{O}}\left((\log n)^{-1/6} + e^{10d^2\sqrt{\log n}}n^{-\frac{1}{d+5}}\right),$$

which tends to 0 as $n$ goes to infinity. $\square$

**Theorem C.5.** *Suppose Assumption 1.1 and Assumption 1.2 hold. Given $n$ samples from target distribution $\pi_1$ and the networks as in Theorem 4.4, with parameter $\zeta$ replaced by $\zeta(\alpha, d)$ defined in Lemma 1.4, we use the estimated velocity field in (11) to generate samples and choose the maximal step size $\max_{k=0,1...,N-1}|t_{k+1} - t_k| = \mathcal{O}(n^{-\frac{4}{3(d+5)}})$ and early stopping time $T(n) = 1 - n^{-\frac{1}{3(d+5)}}$. Then, with probability of at least $1 - \frac{1}{n}$, we have*

$$W_2(\widetilde{\pi}_{T(n)}, \pi_1) = \widetilde{\mathcal{O}}\left(n^{-\frac{1}{3(d+5)}}\right),$$

*where we omit logarithms.*

*Proof.* Lemma 1.4 shows that the velocity field $\boldsymbol{v}^*$ is $\zeta(\alpha, d)$-Lipschitz on $\mathbb{R}^d \times [0, 1]$. The Lipschitz constant only depends on $\alpha$ and dimension $d$. Combining Proposition C.1, Lemma C.2 and Lemma C.3, we obtain

$$W_2(\widetilde{\pi}_{T(n)}, \pi_1) = \widetilde{\mathcal{O}}\left((1 - T) + (K + \gamma_2)\sqrt{\sum_{k=0}^{N-1}(t_{k+1} - t_k)^3 + \frac{1}{(1 - T)^2}n^{-\frac{1}{d+5}}}\right).$$

By letting $\max_{k=0,1\ldots,N-1} |t_{k+1} - t_k| = \mathcal{O}(n^{-\frac{4}{3(d+5)}})$ and $T(n) = 1 - n^{-\frac{1}{3(d+5)}}$, we get the desired result. $\qquad\square$

### C.4. Auxiliary lemma in Appendix C

**Lemma C.6** (Grönwall's inequality). *Given a function $f(t)$ defined on $[a, b]$ ($a < b$), satisfying $\frac{\mathrm{d}f(t)}{\mathrm{d}t} \leq \alpha f(t) + g(t)$ on $[a, b]$ and $\alpha \geq 0$, we have*

$$f(b) \leq e^{\alpha(b-a)}f(a) + \int_a^b e^{\alpha(b-t)}g(t)\mathrm{d}t.$$

*Proof.* By multiplying $e^{-\alpha t}$ on both sides of $\frac{\mathrm{d}f(t)}{\mathrm{d}t} \leq \alpha f(t) + g(t)$ and some manipulation of algebra, we obtain

$$e^{-\alpha t}\frac{\mathrm{d}f(t)}{\mathrm{d}t} - \alpha e^{-\alpha t}f(t) \leq e^{-\alpha t}g(t).$$

Integrating on interval $[a, b]$ on both sides , we get

$$e^{-\alpha b}f(b) - e^{-\alpha a}f(a) \leq \int_a^b e^{\alpha(b-t)}g(t)\mathrm{d}t.$$

This concludes the proof. $\qquad\square$

## D. Properties of true velocity field

### D.1. Computation of true velocity field

**Lemma D.1.** *The true velocity field $\boldsymbol{v}^*$ can be written as:*

$$\boldsymbol{v}^*(\boldsymbol{x}, t) = \frac{1-t}{t}\nabla \log \pi_t(\boldsymbol{x}) + \frac{1}{t}\boldsymbol{x}, \tag{63}$$

*where $\pi_t$ is the density of $X_t$, and $X_t = (1 - t)X_0 + tX_1$.*

*Proof.* By some manipulation of algebra, (3) implies:

$$\begin{aligned}
\boldsymbol{v}^*(\boldsymbol{x}, t) &= \mathbb{E}\left[X_1 - X_0 | X_t = \boldsymbol{x}\right] \\
&= \mathbb{E}\left[X_1 - \frac{1}{1-t}\left((1-t)X_0 + tX_1 - tX_1\right) \Big| X_t = \boldsymbol{x}\right] \\
&= \frac{1}{1-t}\mathbb{E}[X_1 | X_t = \boldsymbol{x}] - \frac{1}{1-t}\boldsymbol{x} \\
&= \frac{1}{1-t}\int \frac{\boldsymbol{x}_1 \pi_{t|1}(\boldsymbol{x}|\boldsymbol{x}_1)\pi_1(\boldsymbol{x}_1)}{\pi_t(\boldsymbol{x})}\mathrm{d}\boldsymbol{x}_1 - \frac{1}{1-t}\boldsymbol{x} \\
&= \frac{1}{1-t}\int \frac{1}{\sqrt{(2\pi)^d(1-t)^{2d}}}\frac{\boldsymbol{x}_1 \exp(-\frac{\|\boldsymbol{x}-t\boldsymbol{x}_1\|^2}{2(1-t)^2})\pi_1(\boldsymbol{x}_1)}{\pi_t(\boldsymbol{x})}\mathrm{d}\boldsymbol{x}_1 - \frac{1}{1-t}\boldsymbol{x} \\
&= \frac{1-t}{t}\int \frac{1}{\sqrt{(2\pi)^d(1-t)^{2d}}}\frac{\left(\frac{t\boldsymbol{x}_1-\boldsymbol{x}}{(1-t)^2} + \frac{\boldsymbol{x}}{(1-t)^2}\right)\exp(-\frac{\|\boldsymbol{x}-t\boldsymbol{x}_1\|^2}{2(1-t)^2})\pi_1(\boldsymbol{x}_1)}{\pi_t(\boldsymbol{x})}\mathrm{d}\boldsymbol{x}_1 - \frac{1}{1-t}\boldsymbol{x} \\
&= \frac{1-t}{t}\int \frac{1}{\sqrt{(2\pi)^d(1-t)^{2d}}}\frac{\nabla_{\boldsymbol{x}}\exp\left(-\frac{\|\boldsymbol{x}-t\boldsymbol{x}_1\|^2}{2(1-t)^2}\right)\pi_1(\boldsymbol{x}_1)}{\pi_t(\boldsymbol{x})}\mathrm{d}\boldsymbol{x}_1 + \left(\frac{1}{t(1-t)} - \frac{1}{1-t}\right)\boldsymbol{x} \\
&= \frac{1-t}{t}\nabla_{\boldsymbol{x}}\log \pi_t(\boldsymbol{x}) + \frac{1}{t}\boldsymbol{x},
\end{aligned}$$

where $\pi_{t|1}$ is the density of $X_t$ conditioned on $X_1$. It concludes the proof. $\qquad\square$

### D.2. Computation of partial derivative regarding $t$

**Lemma D.2.** $\partial_t v^*(\boldsymbol{x}, t) = -\frac{1}{(1-t)^2}\boldsymbol{x} + \frac{1}{(1-t)^2}\mathbb{E}[X_1|X_t = \boldsymbol{x}] + \frac{1+t}{(1-t)^4}\mathrm{Cov}[X_1|X_t = \boldsymbol{x}]\boldsymbol{x} - \frac{t}{(1-t)^4}\left(\mathbb{E}[X_1\|X_1\|^2|X_t = \boldsymbol{x}] - \mathbb{E}[X_1|X_t = \boldsymbol{x}]\mathbb{E}[\|X_1\|^2|X_t = \boldsymbol{x}]\right)$, where $\mathrm{Cov}[X_1|X_t = \boldsymbol{x}]$ is the covariance matrix of $X_1$ conditioned on $X_t = \boldsymbol{x}$.

*Proof.* To ease notation, we define $\phi_t(\boldsymbol{x}) := \int \exp\left(-\frac{\|\boldsymbol{x}-t\boldsymbol{x}_1\|^2}{2(1-t)^2}\right)\pi_1(\mathrm{d}\boldsymbol{x}_1)$, which is the unnormalized version of $\pi_t(\boldsymbol{x})$. Note that $\nabla\log\phi_t(\boldsymbol{x}) = \nabla\log\pi_t(\boldsymbol{x})$, using the product rule of the derivatives, (63) implies:

$$
\begin{aligned}
\partial_t v^*(\boldsymbol{x}, t) &= -\frac{1}{t^2}\nabla\log\pi_t(\boldsymbol{x}) + \frac{1-t}{t}\partial_t\nabla\log\pi_t(\boldsymbol{x}) - \frac{1}{t^2}\boldsymbol{x} \\
&= -\frac{1}{t(1-t)^2}\mathbb{E}[X_1|X_t = \boldsymbol{x}] + \frac{1}{t^2(1-t)^2}\boldsymbol{x} + \frac{1-t}{t}\partial_t\left(\frac{\nabla\phi_t(\boldsymbol{x})}{\phi_t(\boldsymbol{x})}\right) - \frac{1}{t^2}\boldsymbol{x} \\
&= \frac{2-t}{t(1-t)^2}\boldsymbol{x} - \frac{1}{t(1-t)^2}\mathbb{E}[X_1|X_t = \boldsymbol{x}] + \frac{1-t}{t}\left(\frac{\partial_t\nabla\phi_t(\boldsymbol{x})}{\phi_t(\boldsymbol{x})} - \frac{\partial_t\phi_t(\boldsymbol{x})\nabla\phi_t(\boldsymbol{x})}{(\phi_t(\boldsymbol{x}))^2}\right)
\end{aligned}
\tag{64}
$$

Then we focus on the computation of the last term above. We first compute $\frac{\partial_t\nabla\phi_t(\boldsymbol{x})}{\phi_t(\boldsymbol{x})}$ as follows:

$$
\begin{aligned}
\frac{\partial_t\nabla\phi_t(\boldsymbol{x})}{\phi_t(\boldsymbol{x})} &= \frac{1}{\phi_t(\boldsymbol{x})}\partial_t\int\frac{t\boldsymbol{x}_1 - \boldsymbol{x}}{(1-t)^2}\exp\left(-\frac{\|\boldsymbol{x}-t\boldsymbol{x}_1\|^2}{2(1-t)^2}\right)\pi_1(\mathrm{d}\boldsymbol{x}_1) \\
&= \frac{1}{\phi_t(\boldsymbol{x})}\int\left(\frac{(1-t)^2\boldsymbol{x}_1 - 2(t\boldsymbol{x}_1-\boldsymbol{x})(t-1)}{(1-t)^4}\exp\left(-\frac{\|\boldsymbol{x}-t\boldsymbol{x}_1\|^2}{2(1-t)^2}\right) - \right. \\
&\quad \left.\frac{t\boldsymbol{x}_1 - \boldsymbol{x}}{(1-t)^2}\exp\left(-\frac{\|\boldsymbol{x}-t\boldsymbol{x}_1\|^2}{2(1-t)^2}\right)\frac{(t\|\boldsymbol{x}_1\|^2-\boldsymbol{x}_1^T\boldsymbol{x})(1-t)^2 - (t-1)\|\boldsymbol{x}-t\boldsymbol{x}_1\|^2}{(1-t)^4}\right)\pi_1(\mathrm{d}\boldsymbol{x}_1) \\
&= \frac{1+t}{(1-t)^3}\mathbb{E}[X_1|X_t = \boldsymbol{x}] - \frac{2}{(1-t)^3}\boldsymbol{x} - \frac{t^2}{(1-t)^5}\mathbb{E}[X_1\|X_1\|^2|X_t = \boldsymbol{x}] + \\
&\quad \frac{t(1+t)}{(1-t)^5}\mathbb{E}[X_1X_1^T|X_t = \boldsymbol{x}]\boldsymbol{x} - \frac{t}{(1-t)^5}\mathbb{E}[X_1|X_t = \boldsymbol{x}]\|\boldsymbol{x}\|^2 + \\
&\quad \frac{t}{(1-t)^5}\mathbb{E}[\|X_1\|^2|X_t = \boldsymbol{x}]\boldsymbol{x} - \frac{1+t}{(1-t)^5}\mathbb{E}[X_1^T\boldsymbol{x}|X_t = \boldsymbol{x}]\boldsymbol{x} + \frac{\|\boldsymbol{x}\|^2\boldsymbol{x}}{(1-t)^5}
\end{aligned}
\tag{65}
$$

By some calculus, we have

$$
\begin{aligned}
\frac{\partial_t\phi_t(\boldsymbol{x})}{\phi_t(\boldsymbol{x})} &= \frac{1}{\phi_t(\boldsymbol{x})}\int -\frac{(t\|\boldsymbol{x}_1\|^2 - \boldsymbol{x}_1^T\boldsymbol{x})(1-t)^2 + \|\boldsymbol{x}-t\boldsymbol{x}_1\|^2(1-t)}{(1-t)^4}\exp\left(-\frac{\|\boldsymbol{x}-t\boldsymbol{x}_1\|^2}{2(1-t)^2}\right)\pi_1(\mathrm{d}\boldsymbol{x}_1) \\
&= -\frac{t}{(1-t)^3}\mathbb{E}[\|X_1\|^2|X_t = \boldsymbol{x}] + \frac{1+t}{(1-t)^3}\mathbb{E}[X_1^T\boldsymbol{x}|X_t = \boldsymbol{x}] - \frac{\|\boldsymbol{x}\|^2}{(1-t)^3}
\end{aligned}
\tag{66}
$$

and

$$
\begin{aligned}
\frac{\nabla\phi_t(\boldsymbol{x})}{\phi_t(\boldsymbol{x})} &= \frac{1}{\phi_t(\boldsymbol{x})}\int\frac{t\boldsymbol{x}_1 - \boldsymbol{x}}{(1-t)^2}\exp\left(-\frac{\|\boldsymbol{x}-t\boldsymbol{x}_1\|^2}{2(1-t)^2}\right)\pi_1(\mathrm{d}\boldsymbol{x}_1) \\
&= -\frac{\boldsymbol{x}}{(1-t)^2} + \frac{t}{(1-t)^2}\mathbb{E}[X_1|X_t = \boldsymbol{x}].
\end{aligned}
\tag{67}
$$

Combining (64), (65), (66) and (67), we obtain

$$
\begin{aligned}
\partial_t v^*(\boldsymbol{x}, t) &= -\frac{1}{(1-t)^2}\boldsymbol{x} + \frac{1}{(1-t)^2}\mathbb{E}[X_1|X_t = \boldsymbol{x}] + \frac{1+t}{(1-t)^4}\mathrm{Cov}[X_1|X_t = \boldsymbol{x}]\boldsymbol{x} - \\
&\quad \frac{t}{(1-t)^4}\left(\mathbb{E}[X_1\|X_1\|^2|X_t = \boldsymbol{x}] - \mathbb{E}[X_1|X_t = \boldsymbol{x}]\mathbb{E}[\|X_1\|^2|X_t = \boldsymbol{x}]\right).
\end{aligned}
\tag{68}
$$

It concludes the proof. $\qquad\square$

### D.3. An upper bound for velocity field

**Lemma D.3.** $\sup_{t\in[0,T]} \sup_{\boldsymbol{x}\in[-R,R]^d} |v_i^*(\boldsymbol{x},t)| \leq \frac{1+R}{1-T}$.

*Proof.* For the $i$-coordinate, we have $v_i^* = \frac{1}{1-t}\mathbb{E}[X_1^{(i)}|X_t = \boldsymbol{x}] - \frac{1}{1-t}x_i$, where $X_1^{(i)}$ denotes the $i$-coordinate of $X_1$. Note that $\pi_1$ is supported on $[-1,1]^d$, then

$$\sup_{t\in[0,T]} \sup_{\boldsymbol{x}\in[-R,R]^d} |v_i^*(\boldsymbol{x},t)| \leq \frac{1+R}{1-T}.$$

$\square$

### D.4. An upper bound of partial derivative regarding $t$

**Lemma D.4.** $\sup_{t\in[0,T]} \sup_{\boldsymbol{x}\in[-R,R]^d} |\partial_t \boldsymbol{v}^*(\boldsymbol{x},t)| = \mathcal{O}\left(\frac{d^{3/2}(R+1)}{(1-T)^4}\right)$.

*Proof.* From Lemma D.2, we have

$$\|\partial_t \boldsymbol{v}^*(\boldsymbol{x},t)\| \leq \frac{1}{(1-t)^2}\|\boldsymbol{x}\| + \frac{1}{(1-t)^2}\|\mathbb{E}[X_1|X_t = \boldsymbol{x}]\| + \frac{1+t}{(1-t)^4}\|\mathrm{Cov}[X_1|X_t = \boldsymbol{x}]\|_{\mathrm{op}}\|\boldsymbol{x}\| +$$

$$\frac{t}{(1-t)^4}\left(\|\mathbb{E}[X_1\|X_1\|^2|X_t = \boldsymbol{x}]\| + \|\mathbb{E}[X_1|X_t = \boldsymbol{x}]\|\|\mathbb{E}[\|X_1\|^2|X_t = \boldsymbol{x}]\|\right)$$

Note that $\pi_1$ is assumed to be supported on $[0,1]^d$, we have $\|\mathbb{E}[X_1|X_t = \boldsymbol{x}]\| \leq \mathbb{E}[\|X_1\|^2|X_t = \boldsymbol{x}]^{1/2} \leq d^{1/2}$ and $\|\mathbb{E}[X_1\|X_1\|^2|X_t = \boldsymbol{x}]\| \leq \mathbb{E}[\|X_1\|^6|X_t = \boldsymbol{x}]^{1/2} \leq d^{3/2}$. To bound $\|\mathrm{Cov}[X_1|X_t = \boldsymbol{x}]\|_{\mathrm{op}}$, we have the following inequality for any $\boldsymbol{u} \in \mathbb{R}^d$,

$$\boldsymbol{u}^T \mathrm{Cov}[X_1|X_t = \boldsymbol{x}]\boldsymbol{u} = \mathbb{E}[\boldsymbol{u}^T X_1 X_1^T \boldsymbol{u}|X_t = \boldsymbol{x}] - \mathbb{E}[\boldsymbol{u}^T X_1|X_t = \boldsymbol{x}]\mathbb{E}[X_1^T \boldsymbol{u}|X_t = \boldsymbol{x}]$$
$$= \mathbb{E}[(\boldsymbol{u}^T X_1)^2|X_t = \boldsymbol{x}] - \mathbb{E}[\boldsymbol{u}^T X_1|X_t = \boldsymbol{x}]^2$$
$$\leq 2d\|\boldsymbol{u}\|^2$$

Hence we have $\|\mathrm{Cov}[X_1|X_t = \boldsymbol{x}]\|_{\mathrm{op}} \leq 2d$. Using these above inequalities, we have

$$\sup_{t\in[0,T]} \sup_{\boldsymbol{x}\in[-R,R]^d} \|\partial_t \boldsymbol{v}^*(\boldsymbol{x},t)\| \leq \frac{R\sqrt{d}}{(1-T)^2} + \frac{\sqrt{d}}{(1-T)^2} + \frac{1+T}{(1-T)^4}2d^{3/2}R + \frac{2Td^{3/2}}{(1-T)^4}$$

Note that $T < 1$, the above inequality implies $\sup_{t\in[0,T]} \sup_{\boldsymbol{x}\in[-R,R]^d} \|\partial_t \boldsymbol{v}^*(\boldsymbol{x},t)\| = \mathcal{O}\left(\frac{d^{3/2}(R+1)}{(1-T)^4}\right)$. $\square$

### D.5. Lipschitz continuity regarding spatial variable

Following Wibisono & Jog (2018a;b); Mikulincer & Shenfeld (2021; 2022); Chewi & Pooladian (2022); Gao et al. (2024), we deduce the Lipschitz continuity of the velocity field from the properties of the target distribution. We start by presenting the following lemma showing the connection between the Jacobian matrix of the velocity field and the conditional covariance matrix.

**Lemma D.5.** *We have the following identity:*

$$\nabla \boldsymbol{v}^*(x,t) = \frac{t}{(1-t)^3}\mathrm{Cov}[X_1|X_t = x] - \frac{1}{1-t}I_d.$$

*Proof.* By Lemma D.1, we have

$$\nabla \boldsymbol{v}^*(x,t) = \frac{1-t}{t}\nabla^2 \log \pi_t(\boldsymbol{x}) + \frac{1}{t}I_d.$$

Further, the Hessian $\nabla^2 \log \pi_t(\boldsymbol{x})$ can be computed as

$$\nabla^2 \log \pi_t(\boldsymbol{x}) = \nabla \left( \frac{\int_{\mathbb{R}^d} \frac{t\boldsymbol{x}_1 - \boldsymbol{x}}{(1-t)^2} \exp(-\|\boldsymbol{x} - t\boldsymbol{x}_1\|^2/(1-t)^2)\pi_1(\mathrm{d}\boldsymbol{x}_1)}{\int_{\mathbb{R}^d} \exp(-\|\boldsymbol{x} - t\boldsymbol{x}_1\|^2/(1-t)^2)\pi_1(\mathrm{d}\boldsymbol{x}_1)} \right)$$

$$= -\frac{1}{(1-t)^2} I_d + \frac{\int_{\mathbb{R}^d} \left( \frac{t\boldsymbol{x}_1 - \boldsymbol{x}}{(1-t)^2} \right)^{\otimes 2} \exp(-\|\boldsymbol{x} - t\boldsymbol{x}_1\|^2/(1-t)^2)\pi_1(\mathrm{d}\boldsymbol{x}_1)}{\int_{\mathbb{R}^d} \exp(-\|\boldsymbol{x} - t\boldsymbol{x}_1\|^2/(1-t)^2)\pi_1(\mathrm{d}\boldsymbol{x}_1)}$$

$$- \left( \frac{\int_{\mathbb{R}^d} \frac{t\boldsymbol{x}_1 - \boldsymbol{x}}{(1-t)^2} \exp(-\|\boldsymbol{x} - t\boldsymbol{x}_1\|^2/(1-t)^2)\pi_1(\mathrm{d}\boldsymbol{x}_1)}{\int_{\mathbb{R}^d} \exp(-\|\boldsymbol{x} - t\boldsymbol{x}_1\|^2/(1-t)^2)\pi_1(\mathrm{d}\boldsymbol{x}_1)} \right)^{\otimes 2}$$

$$= -\frac{1}{(1-t)^2} I_d + \frac{t^2}{(1-t)^4} \mathrm{Cov}[X_1|X_t = x].$$

Combing the above identities, we get the desired result. $\qquad \square$

**Lemma D.6.** *Suppose that Assumption 1.1 holds. Then $\boldsymbol{v}^*(\boldsymbol{x}, t)$ is $\xi$-Lipschitz continuous w.r.t. $\boldsymbol{x}$ on $\mathbb{R}^d \times [0, T]$, where $\xi \leq \max\left\{ \frac{1}{1-T}, \frac{Td}{(1-T)^3} \right\}$. Further, if $\frac{1}{2} < T < 1$, we have $\boldsymbol{v}^*$ is $\frac{d}{(1-T)^3}$-Lipschitz continuous w.r.t. $\boldsymbol{x}$.*

*Proof.* Since we assume the target distribution $\pi_1$ is supported on $[0, 1]^d$, we have the following evaluation of the covariance matrix

$$0 \preceq \mathrm{Cov}[X_1|X_t = x] \preceq dI_d.$$

Thus, we have

$$-\frac{1}{1-t} I_d \preceq \nabla \boldsymbol{v}^*(x, t) \preceq \left( \frac{td}{(1-t)^3} - \frac{1}{1-t} \right) I_d.$$

The above inequality implies the Lipschitz constant of $\boldsymbol{v}^*$ w.r.t. $\boldsymbol{x}$. $\qquad \square$

We further need the following two functional inequalities to control the conditional covariance under Assumption 1.2, namely the Brascamp-Lieb inequality (BLI) and Cramér-Rao inequality (CRI).

**Lemma D.7** (Brascamp-Leib inequality). *Let $\mu(\mathrm{d}\boldsymbol{x}) = \exp(-U(\boldsymbol{x}))\mathrm{d}\boldsymbol{x}$ be a probability measure on a convex set $\Omega \subseteq \mathbb{R}^d$ whose potential $U : \Omega \to \mathbb{R}$ is twice continuously differentiable and strictly convex. Then*

$$\mathrm{Cov}_\mu(X) \preceq \mathbb{E}_\mu[(\nabla^2 U(X))^{-1}],$$

*with equality if $X \sim \mathcal{N}(m, \Sigma)$ with $\Sigma$ positive definite.*

The complete proof of BLI can be found in (Brascamp & Lieb, 1976, Theorem 4.1) and (Saumard & Wellner, 2014).

**Lemma D.8** (Cramér-Rao inequality). *Let $\mu(\mathrm{d}\boldsymbol{x}) = \exp(-U(\boldsymbol{x}))\mathrm{d}\boldsymbol{x}$ be a probability measure on a convex set $\Omega \subseteq \mathbb{R}^d$ whose potential $U : \Omega \to \mathbb{R}$ is twice continuously differentiable. Then*

$$\mathrm{Cov}_\mu(X) \succeq (\mathbb{E}_\mu[\nabla^2 U])^{-1},$$

*with equality if $X \sim \mathcal{N}(m, \Sigma)$ with $\Sigma$ positive definite.*

The complete proof of CRI can be found in (Saumard & Wellner, 2014; Dembo et al., 1991).

**Lemma D.9.** *Suppose that Assumption 1.1 and Assumption 1.2 hold. Then $\boldsymbol{v}^*(\boldsymbol{x}, t)$ is $\zeta(\alpha, d)$-Lipschitz continuous on $\mathbb{R}^d \times [0, 1]$ w.r.t. $\boldsymbol{x}$, where $\zeta(\alpha, d)$ scales polynomially with $\alpha$ and $d$.*

*Proof.* Note that

$$-\nabla^2_{\boldsymbol{x}_1} \log \pi_{1|t}(\boldsymbol{x}_1|\boldsymbol{x}) = -\nabla^2_{\boldsymbol{x}_1} \log \pi_1(\boldsymbol{x}_1) - \nabla^2_{\boldsymbol{x}_1} \log \pi_{t|1}(\boldsymbol{x}|\boldsymbol{x}_1),$$

where $\pi_{1|t}$ is the conditional density of $X_1$ conditioned on $X_t = \boldsymbol{x}$ and $\pi_{t|1}$ is the conditional density of $X_t$ conditioned on $X_1 = \boldsymbol{x}_1$. Since $X_t$ can be viewed as $tX_1$ perturbed by a Gaussian noise, we have $\pi_{t|1}(\boldsymbol{x}|\boldsymbol{x}_1) \propto \exp\left( -\frac{\|\boldsymbol{x} - t\boldsymbol{x}_1\|^2}{(1-t)^2} \right)$. Thus, we obtain

$$-\nabla^2_{\boldsymbol{x}_1} \log \pi_{1|t}(\boldsymbol{x}_1|\boldsymbol{x}) = -\nabla^2_{\boldsymbol{x}_1} \log \pi_1(\boldsymbol{x}_1) + \frac{t^2}{(1-t)^2} I_d.$$

Assumption 1.2 implies

$$\left(-\alpha + \frac{t^2}{(1-t)^2}\right) I_d \preceq -\nabla^2_{\boldsymbol{x}_1} \log \pi_{1|t}(\boldsymbol{x}_1|\boldsymbol{x}) \preceq \left(\alpha + \frac{t^2}{(1-t)^2}\right) I_d.$$

By the Cramér-Rao inequality, we obtain

$$\text{Cov}[X_1|X_t = \boldsymbol{x}] \succeq \left(\alpha + \frac{t^2}{(1-t)^2}\right)^{-1} I_d. \tag{69}$$

When $t \in \left\{t \in (0,1) : -\alpha + \frac{t^2}{(1-t)^2} > 0\right\}$, by Brascamp-Lieb inequality, we obtain

$$\text{Cov}[X_1|X_t = \boldsymbol{x}] \preceq \left(-\alpha + \frac{t^2}{(1-t)^2}\right)^{-1} I_d. \tag{70}$$

Combining (69) and Lemma D.5, we have

$$\nabla \boldsymbol{v}^*(\boldsymbol{x}, t) \succeq \frac{t - \alpha(1-t)}{\alpha(1-t)^2 + t^2} I_d.$$

Combining (70) and Lemma D.5, for $t \in \left\{t \in (0,1) : -\alpha + \frac{t^2}{(1-t)^2} > 0\right\}$, we have

$$\nabla \boldsymbol{v}^*(\boldsymbol{x}, t) \preceq \frac{t + \alpha(1-t)}{-\alpha(1-t)^2 + t^2} I_d. \tag{71}$$

Recalling the result in Lemma D.6, we have

$$-\frac{1}{1-t} I_d \preceq \nabla \boldsymbol{v}^*(x, t) \preceq \left(\frac{td}{(1-t)^3} - \frac{1}{1-t}\right) I_d.$$

By some manipulation of algebra, it is obvious that $-\frac{1}{1-t} \leq \frac{t-\alpha(1-t)}{\alpha(1-t)^2+t^2}$. Thus, we have

$$\nabla \boldsymbol{v}^*(x, t) \succeq \frac{t - \alpha(1-t)}{\alpha(1-t)^2 + t^2} I_d \succeq \frac{-\alpha}{\alpha/(1+\alpha)} I_d = -(1+\alpha)I_d,$$

where the second inequality follows from the fact that $t - \alpha(1-t) \geq -\alpha$ on $t \in (0,1)$ and $\alpha(1-t)^2 + t^2 \geq \frac{\alpha}{1+\alpha}$. Next, we compare $\frac{td}{(1-t)^3} - \frac{1}{1-t}$ and $\frac{1}{1-t} \cdot \frac{t}{-\alpha(1-t)^2+t^2} - \frac{1}{1-t}$. Let the two quantities be equal, we obtain

$$\frac{2}{d} + \alpha = \frac{t^2}{(1-t)^2}.$$

The root of the above equality in $(0,1)$ is $\frac{\sqrt{\alpha + \frac{2}{d}}}{1 + \sqrt{\alpha + \frac{2}{d}}}$. By the monotonicity of $\frac{x}{1+x}$ on $(0,1)$, we have $\frac{\sqrt{\alpha + \frac{2}{d}}}{1 + \sqrt{\alpha + \frac{2}{d}}} > \frac{\sqrt{\alpha}}{1 + \sqrt{\alpha}}$. Based on this discussion, we obtain

$$\nabla \boldsymbol{v}^*(x, t) \preceq g(t) I_d,$$

where

$$g(t) = \begin{cases} \left(\dfrac{td}{(1-t)^3} - \dfrac{1}{1-t}\right), & t \in \left(0, \dfrac{\sqrt{\alpha + \frac{2}{d}}}{1 + \sqrt{\alpha + \frac{2}{d}}}\right) \\[3ex] \dfrac{t + \alpha(1-t)}{-\alpha(1-t)^2 + t^2}, & t \in \left(\dfrac{\sqrt{\alpha + \frac{2}{d}}}{1 + \sqrt{\alpha + \frac{2}{d}}}, 1\right). \end{cases}$$

By taking the derivative of $\frac{td}{(1-t)^3} - \frac{1}{1-t}$, we can see that $\frac{td}{(1-t)^3} - \frac{1}{1-t}$ is increasing on $\left( 0, \frac{\sqrt{\alpha + \frac{2}{d}}}{1 + \sqrt{\alpha + \frac{2}{d}}} \right)$. Using the same argument, it can be shown that $\frac{t + \alpha(1-t)}{-\alpha(1-t)^2 + t^2}$ is decreasing on $\left( \frac{\sqrt{\alpha + \frac{2}{d}}}{1 + \sqrt{\alpha + \frac{2}{d}}}, 1 \right)$. Based on the above discussion, we obtain

$$\nabla \boldsymbol{v}^*(x, t) \preceq \frac{d}{2} \left( \alpha + \sqrt{\alpha + \frac{2}{d}} \right)^2 I_d.$$

$\square$

