# OpenReview forum: "An Error Analysis of Flow Matching for Deep Generative Modeling"
_ICML.cc/2025/Conference — ICML 2025 spotlightposter_

### Official Review · Reviewer_LksU · 2025-03-10

**Overall Recommendation:** 4

**Summary:**

This paper presents the first end-to-end analysis of Continuous Normalizing Flows (CNFs) built upon Flow Matching. The theoretical results demonstrate that the generated distribution is guaranteed to converge to the true distribution under a mild assumption. Furthermore, the convergence rate is significantly improved assuming a mild Lipschitz condition on the target score function.、
## update after rebuttal
My assessment has not changed. The authors have successfully addressed the raised issues, and my recommendation remains as an "Accept"

**Claims And Evidence:**

All the claims are supported by the theoretical results in this paper.

**Essential References Not Discussed:**

I find the literature review comprehensive.

**Experimental Designs Or Analyses:**

There is no experiments in this paper.

**Methods And Evaluation Criteria:**

There is no experiments in this paper.

**Other Comments Or Suggestions:**

None.

**Other Strengths And Weaknesses:**

This paper relaxes the strong assumptions on the underlying velocity field in the previous work.

**Questions For Authors:**

Is assuming early stopping equivalent to considering $\sigma_{min}$ as the original FM paper? Can the analysis be simplified for some predefined small $\sigma_{min}$ where we resort to a noisy approximation of the target distribution?

**Relation To Broader Scientific Literature:**

The consistency of FM is mainly based on a mild assumption, i.e. boundedness, which justifies the use of CNFs based on FM. Theorem 1.7 highlight the effectiveness of CNFs based on FM in learning the underlying smooth distribution.

**Theoretical Claims:**

I checked the proof for the theoretical claims and did not find any mistake.

---

> ### Author Rebuttal · Authors · 2025-03-30
>
> Thank you for your positive evaluation.
>
> **1. Is assuming early stopping $\sigma_{min}$ equivalent to considering as the original FM paper? Can the analysis be simplified for some predefined small $\sigma_{min}$ where we resort to a noisy approximation of the target distribution?**
>
> **A:** Thank you for providing the insightful question. Your clarification makes sense. The analysis can be simplified when the stopping time is pre-defined. However, as we would prefer the generated distribution to converge to the target distribution, we let the stopping time converge to $0$ as the size of samples increases.

---

### Official Review · Reviewer_8qqb · 2025-03-11

**Overall Recommendation:** 4

**Summary:**

This paper presents an analysis of Continuous Normalizing Flows (CNFs) built upon Flow Matching (FM) for deep generative modeling. It proves the generated distribution of FM converges to the target distribution in the Wasserstein-2 distance for general target distributions with bounded support. The convergence rate is significantly improved under a mild Lipschitz condition of the target score function.

**Claims And Evidence:**

The claims made in the paper appear to be well-supported by theoretical analysis, mathematical proofs, and assumptions outlined throughout the text. Below is an evaluation of the main claims and their supporting evidence:

1. End-to-End Error Analysis: Theorem 1.6, 1.7.

2. Improved Convergence Rate Under Lipschitz Conditions: Theorem 1.7.

**Essential References Not Discussed:**

None.

**Experimental Designs Or Analyses:**

None.

**Methods And Evaluation Criteria:**

Yes.

**Other Comments Or Suggestions:**

None.

**Other Strengths And Weaknesses:**

Strengths:
1) It presents itself as the first end-to-end error analysis of CNFs using FM. This novelty is significant, as it fills a gap in the existing literature regarding the theoretical underpinnings of FM and its implications for generative models.
2) The authors focus on mild assumptions, such as bounded support and Lipschitz conditions, which increases the applicability of their findings to a wider range of practical situations. This accessibility is a strength as it allows for broader implications in real-world applications.

Weaknesses:

1）There is no experiments to support the theoretical results.

**Questions For Authors:**

1) In your analysis, you emphasize the importance of the Lipschitz continuity of the velocity field. Can you elaborate on how varying the Lipschitz constant impacts the convergence rate and the overall performance of the model?
2) What are the most pressing open questions or future research directions that you believe need to be addressed to further advance the understanding and applicability of FM in generative modeling?

**Relation To Broader Scientific Literature:**

FM is highlighted as a pivotal development that enhances the training and efficiency of CNFs. Prior works on FM and interpolated transport paths, notably by Liu et al. (2023) and Karras et al. (2022), are referenced, indicating that this paper builds on and extends these concepts. This situates the work within a lineage of research that seeks to improve existing flow models by addressing their sampling efficiency.

**Theoretical Claims:**

I have skimmed the proofs and believe they are correct.

---

> ### Author Rebuttal · Authors · 2025-03-30
>
> Thank you for your positive evaluation and insightful questions.
>
> **1. There is no experiments to support the theoretical results.**
>
> **A:** We appreciate the reviewer’s concern regarding the absence of experiments. Our primary focus in this work is to establish a rigorous theoretical foundation for Flow Matching (FM), providing theoretical guarantees for its effectiveness.
>
> **2. In your analysis, you emphasize the importance of the Lipschitz continuity of the velocity field. Can you elaborate on how varying the Lipschitz constant impacts the convergence rate and the overall performance of the model?**
>
> **A:** As stated in Theorem 4.1, an increase in the Lipschitz constant leads to greater neural network complexity. This, in turn, expands the hypothesis class, making estimation more challenging and potentially impacting the convergence rate.
>
> **3. What are the most pressing open questions or future research directions that you believe need to be addressed to further advance the understanding and applicability of FM in generative modeling?**
>
> **A:** One crucial direction is to relax the current assumptions while maintaining convergence guarantees, thereby broadening the applicability of FM.

---

### Official Review · Reviewer_C3LC · 2025-03-13

**Overall Recommendation:** 4

**Summary:**

This paper presents the first comprehensive analysis of Continuous Normalizing Flows (CNFs) based on Flow Matching. The theoretical results establish that the generated distribution converges to the true distribution under a mild assumption. Additionally, the convergence rate is notably improved when a mild Lipschitz condition is assumed on the target score function.

**Claims And Evidence:**

Yes

**Essential References Not Discussed:**

There are some references needed to be cited:

[1] Gat I, Remez T, Shaul N, et al. Discrete flow matching. NIPS, 2025.
[2] Shi Y, De Bortoli V, Campbell A, et al. Diffusion Schrödinger bridge matching. NIPS, 2024.
[3] Klein L, Krämer A, Noé F. Equivariant flow matching. NIPS, 2024.

**Experimental Designs Or Analyses:**

NA

**Methods And Evaluation Criteria:**

Yes

**Other Comments Or Suggestions:**

No

**Other Strengths And Weaknesses:**

Strength 1： The paper is well-structured and easy to follow. The authors have conducted a thorough literature review, covering key works on Flow Matching and Diffusion, including some of the most recent advancements in the field.
Strength 2: The results presented in the paper include consistency, generalization bounds, and sample complexity bounds, which comprehensively address the key theoretical questions related to this line of methods. Notably, most of the results are derived under mild assumptions, making the findings both robust and broadly applicable.

Weakness 1: The results depend on specific assumptions (bounded support, Lipschitz continuity) which may not hold in all practical scenarios.

**Questions For Authors:**

No

**Relation To Broader Scientific Literature:**

These results not only enrich the toolbox for theoretical analysis for Flow Matching but also provide new perspectives and methods for practical applications.

**Theoretical Claims:**

I have reviewed the proof for Theorems 1.6 and 1.7 and find no mistakes.

---

> ### Author Rebuttal · Authors · 2025-03-30
>
> Thank you for your positive evaluation.
>
> **1. The results depend on specific assumptions (bounded support, Lipschitz continuity) which may not hold in all practical scenarios.**
>
> **A:** We appreciate the reviewer’s insightful comments on the assumptions of bounded support and Lipschitz continuity. These conditions are standard in the literature, but we acknowledge the importance of exploring more general settings. In future work, we aim to extend our analysis to relax these assumptions and investigate their impact on our results.

---

### Official Review · Reviewer_qBs8 · 2025-03-13

**Overall Recommendation:** 3

**Summary:**

This paper provides an analysis of flow matching. The authors prove that generative models based on flow matching converge to the target distribution under mild assumption.

## update after rebuttal ##
I have reviewed the rebuttal and decided to maintain the original score.

**Claims And Evidence:**

I'm very unfamilar with this domain, although I tried my best to understand its content and provide thoughtful reviews.

**Essential References Not Discussed:**

I did not identify any major related works that are missing.

**Experimental Designs Or Analyses:**

This is a theory-related paper, with no experiments provided.

**Methods And Evaluation Criteria:**

The paper does not propose specific methods or evaluation criteria, as it is a theoretical contribution. The validity of the results depends on the correctness and rigor of the proofs rather than empirical benchmarks.

**Other Comments Or Suggestions:**

The introduction is too brief. Many existing works have analyzed FM, and the introduction should clearly explain how this work differs from previous theory-related works.
The authors should provide an overview at the end of the introduction. Currently, the transition from the main contributions  directly to "1.1 Assumptions" is abrupt, making it unclear what this part aims to achieve.
Both Section 2 and Section 3 are titled "Preliminaries", overly confusing. Would it not be more logical to merge them into a single section?

**Other Strengths And Weaknesses:**

N/A

**Questions For Authors:**

Please address my concerns or correct me if there is anything wrong in other comments.

**Relation To Broader Scientific Literature:**

The key contributions of this paper are closely related to the broader scientific literature in generative modeling.

**Theoretical Claims:**

The discussion of the theoretical aspects in the paper appears to be reasonable.

---

> ### Author Rebuttal · Authors · 2025-03-30
>
> Thank you for your valuable suggestions.
>
> **1. Many existing works have analyzed FM, and the introduction should clearly explain how this work differs from previous theory-related works.**
>
> **A:** While recent works [1,2,3] have analyzed ODE-based generative models, they typically assume that the score function or velocity function is uniformly Lipschitz over all time steps. In contrast, our analysis drops these assumptions, which makes the analysis more general.
>
> **2. The authors should provide an overview at the end of the introduction.**
>
> **A:** Here is a brief overview, which we will include in our revision. In Preliminaries, we introduce the necessary notations and background on Flow Matching (FM) and Continuous Normalizing Flows (CNFs). Section 4 analyzes the approximation error that arises when using neural networks to approximate the true velocity field. Section 5 studies the estimation error in learning the velocity field from data. Section 6 studies the discretization error introduced when solving the ODE flow numerically.
>
> **3. Currently, the transition from the main contributions directly to "1.1 Assumptions" is abrupt, ... Would it not be more logical to merge them into a single section?**
>
> **A:** We agree that the transition is currently abrupt and will revise the structure to ensure the readability. Specifically, we will either integrate the key assumptions into the contributions section or provide a brief motivation before introducing them.
>
>
>
>
> [1] Albergo, M. S., Boffi, N. M., and Vanden-Eijnden, E. Stochastic interpolants: A unifying framework for flows and diffusions.
> [2] Chen, S., Daras, G., and Dimakis, A. G. Restoration-degradation beyond linear diffusions: A non-asymptotic analysis for ddim-type samplers.
> [3] Lu, C., Zheng, K., Bao, F., Chen, J., Li, C., and Zhu, J. Maximum likelihood training for score-based diffusion odes by high order denoising score matching.

---

### Decision · Program_Chairs · 2025-05-01

**Decision:**

Accept (spotlight poster)

**Comment:**

After the rebuttal and discussion phases, the paper received scores of 4, 4, 4, and 3, which exceed the expected threshold for acceptance. After briefly reviewing the comments and the authors' responses, I believe the paper meets the acceptance criteria for ICML.